# GlucoBench: Curated List of Continuous Glucose Monitoring Datasets with Prediction Benchmarks

**Renat Sergazinov**[1]*, **Elizabeth Chun**[1], **Valeriya Rogovchenko**[1],
**Nathaniel Fernandes**[2], **Nicholas Kasman**[2], **Irina Gaynanova**[1]*
[1]Department of Statistics, Texas A&M University
[2]Department of Electrical and Computer Engineering, Texas A&M University

## Abstract

The rising rates of diabetes necessitate innovative methods for its management. Continuous glucose monitors (CGM) are small medical devices that measure blood glucose levels at regular intervals providing insights into daily patterns of glucose variation. Forecasting of glucose trajectories based on CGM data holds the potential to substantially improve diabetes management, by both refining artificial pancreas systems and enabling individuals to make adjustments based on predictions to maintain optimal glycemic range. Despite numerous methods proposed for CGM-based glucose trajectory prediction, these methods are typically evaluated on small, private datasets, impeding reproducibility, further research, and practical adoption. The absence of standardized prediction tasks and systematic comparisons between methods has led to uncoordinated research efforts, obstructing the identification of optimal tools for tackling specific challenges. As a result, only a limited number of prediction methods have been implemented in clinical practice.

To address these challenges, we present a comprehensive resource that provides (1) a consolidated repository of curated publicly available CGM datasets to foster reproducibility and accessibility; (2) a standardized task list to unify research objectives and facilitate coordinated efforts; (3) a set of benchmark models with established baseline performance, enabling the research community to objectively gauge new methods' efficacy; and (4) a detailed analysis of performance-influencing factors for model development. We anticipate these resources to propel collaborative research endeavors in the critical domain of CGM-based glucose predictions. Our code is available online at github.com/IrinaStatsLab/GlucoBench.

## 1 Introduction

According to the International Diabetes Federation, 463 million adults worldwide have diabetes with 34.2 million people affected in the United States alone (IDF, 2021). Diabetes is a leading cause of heart disease (Nanayakkara et al., 2021), blindness (Wykoff et al., 2021), and kidney disease (Alicic et al., 2017). Glucose management is a critical component of diabetes care, however achieving target glucose levels is difficult due to multiple factors that affect glucose fluctuations, e.g., diet, exercise, stress, medications, and individual physiological variations.

Continuous glucose monitors (CGM) are medical devices that measure blood glucose levels at frequent intervals, often with a granularity of approximately one minute. CGMs have great potential to improve diabetes management by furnishing real-time feedback to patients and by enabling an autonomous artificial pancreas (AP) system when paired with an insulin pump (Contreras & Vehi, 2018; Kim & Yoon, 2020). Figure 1 illustrates an example of a CGM-human feedback loop in a recommender setting. The full realization of CGM potential, however, requires accurate glucose prediction models. Although numerous prediction models (Fox et al., 2018; Armandpour et al., 2021; Sergazinov et al., 2023) have been proposed, only simple physiological (Bergman et al., 1979; Hovorka et al., 2004) or statistical (Oviedo et al., 2017; Mirshekarian et al., 2019; Xie & Wang,

---

*Address correspondence to: mrsergazinov@tamu.edu, irinag@tamu.edu

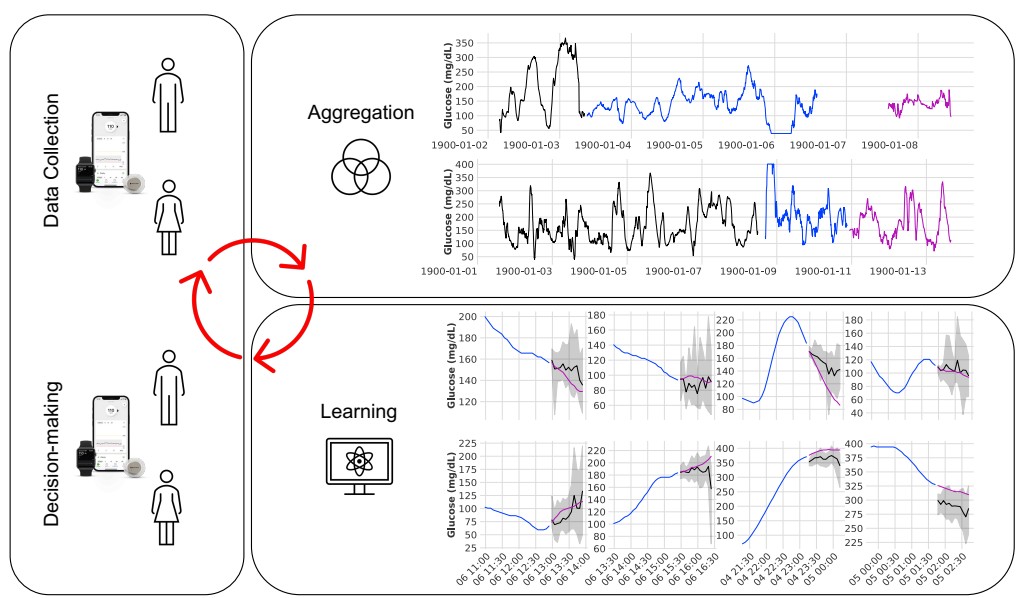

Figure 1: Sample of glucose curves captured by the Dexcom G4 Continuous Glucose Monitoring (CGM) system, with dates de-identified for privacy(Weinstock et al., 2016).

2020) models are utilized within current CGM and AP software. The absence of systematic model evaluation protocols and established benchmark datasets hinder the analysis of more complex models' risks and benefits, leading to their limited practical adoption (Mirshekarian et al., 2019).

In response, we present a curated list of five public CGM datasets and a systematic protocol for models' evaluation and benchmarking. The selected datasets have varying sizes and demographic characteristics, while the developed systematic data-preprocessing pipeline facilitates the inclusion of additional datasets. We propose two tasks: (1) enhancing the predictive accuracy; (2) improving the uncertainty quantification (distributional fit) associated with predictions. In line with previous works (Mirshekarian et al., 2019; Xie & Wang, 2020), we measure the performance on the first task with mean squared error (MSE) and mean absolute error (MAE), and on the second task with likelihood and expected calibration error (ECE) (Kuleshov et al., 2018). For each task, we train and evaluate a set of baseline models. From data-driven models, we select linear regression and ARIMA to represent shallow baselines, and Transformer (Vaswani et al., 2017), NHiTS (Challu et al., 2023), TFT (Lim et al., 2021), and Gluformer (Sergazinov et al., 2023) to represent deep learning baselines. We select the Latent ODE (Rubanova et al., 2019) to represent a hybrid data-driven / physiological model.

Our work contributes a **curated collection** of diverse CGM datasets, **formulation of two tasks** focused on model accuracy and uncertainty quantification, an efficient **benchmarking protocol**, evaluation of a range of **baseline models** including shallow, deep and hybrid methods, and a **detailed analysis of performance-influencing factors** for model optimization.

## 2 RELATED WORKS

An extensive review of glucose prediction models and their utility is provided by Oviedo et al. (2017); Contreras & Vehi (2018); Kim & Yoon (2020); Kavakiotis et al. (2017). Following Oviedo et al. (2017), we categorize prediction models as physiological, data-driven, or hybrid. **Physiological models** rely on the mathematical formulation of the dynamics of insulin and glucose metabolism via differential equations (Man et al., 2014; Lehmann & Deutsch, 1991). A significant limitation of these models is the necessity to pre-define numerous parameters. **Data-driven models** rely solely on the CGM data (and additional covariates when available) to characterize glucose trajectory without incorporating physiological dynamics. These models can be further subdivided into shallow (e.g. linear regression, ARIMA, random forest, etc.) and deep learning models (e.g. recurrent neural network models, Transformer, etc.). Data-driven models, despite their capacity to capture complex

Table 1: Summary of the glucose prediction models by dataset and model type. We indicate "open" for datasets that are publicly available online, "simulation" for the ones based on simulated data, "proprietary" for the ones that cannot be released. We indicate deep learning models by "deep", non-deep learning models by "shallow", and physiological models by "physiological." We provide full table with references to all the works in Appendix A.

| Type | Diabetes | # of datasets | # of deep | # of shallow | # of Physiological |
|---|---|---|---|---|---|
| Open | Type 1 | 9 | 13 | 3 | 2 |
| Simulation | Type 1 | 12 | 3 | 3 | 6 |
| Proprietary | Mixed | 22 | 7 | 8 | 7 |

patterns, may suffer from overfitting and lack interpretability. **Hybrid models** use physiological models as a pre-processing or data augmentation tool for data-driven models. Hybrid models enhance the flexibility of physiological models and facilitate the fitting process, albeit at the expense of diminished interpretability. Table 1 summarizes existing models and datasets, indicating model type.

**Limitations.** The present state of the field is characterized by several key constraints, including (1) an absence of well-defined benchmark datasets and studies, (2) a dearth of open-source code bases, and (3) omission of Type 2 diabetes from most open CGM studies. To address the second limitation, two benchmark studies have been undertaken to assess the predictive performance of various models (Mirshekarian et al., 2019; Xie & Wang, 2020). Nonetheless, these studies only evaluated the models on one dataset (Marling & Bunescu, 2020), comprising a limited sample of 5 patients with Type 1 diabetes, and failed to provide source code. We emphasize that, among the 45 methods identified in Table 1, a staggering 38 works do not offer publicly available implementations. For the limitation (3), it is important to recognize that Type 2 is more easily managed through lifestyle change and oral medications than Type 1 which requires lifelong insulin therapy.

## 3 DATA

### 3.1 DESCRIPTION

We have selected five publicly available CGM datasets: Broll et al. (2021); Colás et al. (2019); Dubosson et al. (2018); Hall et al. (2018); Weinstock et al. (2016).

To ensure data quality, we used the following set of criteria. First, we included a variety of dataset sizes and verified that each dataset has measurements on at least 5 subjects and that the collection includes a variety of sizes ranging from only five (Broll et al., 2021) to over 200 (Colás et al., 2019; Weinstock et al., 2016) patients. On the patient level, we ensured that each subject has non-missing CGM measurements for at least 16 consecutive hours. At the CGM curve level, we have verified that measurements fall within a clinically relevant range of 20 mg/dL to 400 mg/dL, avoiding drastic fluctuations exceeding 40 mg/dL within a 5-minute interval, and ensuring non-constant values.

Finally, we ensured that the collection covers distinct population groups representing subjects with Type 1 diabetes (Dubosson et al., 2018; Weinstock et al., 2016), Type 2 diabetes (Broll et al., 2021), or a mix of Type 2 and none (Colás et al., 2019; Hall et al., 2018). We expect that the difficulty of accurate predictions will depend on the population group: patients with Type 1 have significantly larger and more frequent glucose fluctuatons. Table 2 summarizes all five datasets with associated demographic information, where some subjects are removed due to data quality issues as a result of pre-processing (Section 3.2). We describe data availability in Appendix A.

**Covariates.** In addition to CGM data, each dataset has covariates (features), which we categorize based on their temporal structure and input type. The temporal structure distinguishes covariates as static (e.g. gender), dynamic known (e.g. hour, minute), and dynamic unknown (e.g. heart beat, blood pressure). Furthermore, input types define covariates as either real-valued (e.g. age) or ordinal (e.g. education level) and categorical or unordered (e.g. gender) variables. We illustrate different types of temporal variables in Figure 2. We summarize covariate types for each dataset in Appendix A.

Table 2: Demographic information (average) for each dataset before (Raw) and after pre-processing (Processed). CGM indicates the device type; all devices have 5 minute measurment frequency.

| Dataset | Diabetes | CGM | # of Subjects | | Age | | Sex (M / F) | |
|---|---|---|---|---|---|---|---|---|
| | Overall | Overall | Raw | Processed | Raw | Processed | Raw | Processed |
| Broll et al. (2021) | Type 2 | Dexcom G4 | 5 | 5 | NA | NA | NA | NA |
| Colás et al. (2019) | Mixed | MiniMed iPro | 208 | 201 | 59 | 59 | 103 / 104 | 100 / 100 |
| Dubosson et al. (2018) | Type 1 | MiniMed iPro2 | 9 | 7 | NA | NA | 6 / 3 | NA |
| Hall et al. (2018) | Mixed | Dexcom G4 | 57 | 56 | 48 | 48 | 25 / 32 | NA |
| Weinstock et al. (2016) | Type 1 | Dexcom G4 | 200 | 192 | 68 | NA | 106 / 94 | 101 / 91 |

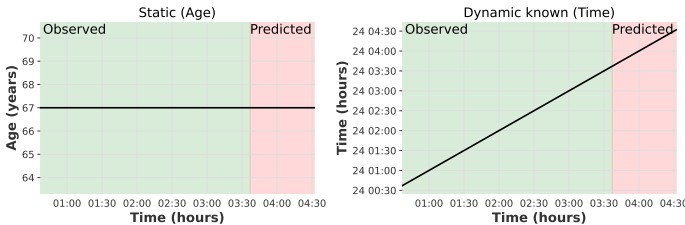 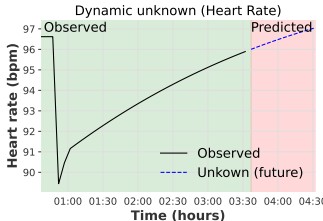

Figure 2: An illustration of static (Age), dynamic known (Date), and dynamic unknown (Heart Rate) covariate categories based on data from Hall et al. (2018) and Dubosson et al. (2018).

Table 3: Interpolation parameters for datasets.

| Parameters | Broll | Colas | Dubosson | Hall | Weinstock |
|---|---|---|---|---|---|
| Gap threshold (minutes) | 45 | 45 | 30 | 30 | 45 |
| Minimum length (hours) | 20 | 16 | 20 | 16 | 20 |

## 3.2 PRE-PROCESSING

We pre-process the raw CGM data via interpolation and segmentation, encoding categorical features, data partitioning, scaling, and division into input-output pairs for training a predictive model.

**Interpolation and segmentation.** To put glucose values on a uniform temporal grid, we identify gaps in each subject's trajectory due to missing measurements. When the gap length is less than a predetermined threshold (Table 3), we impute the missing values by linear interpolation. When the gap length exceeds the threshold, we break the trajectory into several continuous segments. Green squares in Figure 3 indicate gaps that will be interpolated, whereas red arrows indicate larger gaps where the data will be broken into separate segments. In Dubosson et al. (2018) dataset, we also interpolate missing values in dynamic covariates (e.g., heart rate). Thus, for each dataset we obtain a list of CGM sequences $\mathcal{D} = \{\mathbf{x}_j^{(i)}\}_{i,j}$ with $i$ indexing the patients and $j$ the continuous segments. Each segment $\mathbf{x}_j^{(i)}$ has length $L_j^{(i)} > L_{min}$, where $L_{min}$ is the pre-specified minimal value (Table 3).

**Covariates Encoding.** While many of the covariates are real-valued, e.g., age, some covariates are categorical, e.g., sex. In particular, Weinstock et al. (2016) dataset has 36 categorical covariates with an average of 10 levels per covariate. While one-hot encoding is a popular approach for modeling categorical covariates, it will lead to 360 feature columns on Weinstock et al. (2016), making it undesirable from model training time and memory perspectives. Instead, we use label encoding by converting each categorical level into a numerical value. Given $R$ covariates, we include them in the dataset as $\mathcal{D} = \{\mathbf{x}_j^{(i)}, \mathbf{c}_{1,j}^{(i)}, \dots \mathbf{c}_{R,j}^{(i)}\}_{i,j}$ where $\mathbf{c}_{r,j}^{(i)} \in \mathbb{R}$ for static and $\mathbf{c}_{r,j}^{(i)} \in \mathbb{R}^{L_j^{(i)}}$ for dynamic.

**Data splitting.** Each dataset is split into train, validation, and in-distribution (ID) test sets using 90% of subjects. For each subject, the sets follow chronological time order as shown in Figure 3, with validation and ID test sets always being of a fixed length of 16 hours each (192 measurements). The data from the remaining 10% of subjects is used to form an out-of-distribution (OD) test set to assess the generalization abilities of predictive models as in Section 5.2. Thus, $\mathcal{D} = \mathcal{D}_{tr} \cup \mathcal{D}_{val} \cup \mathcal{D}_{id} \cup \mathcal{D}_{od}$.

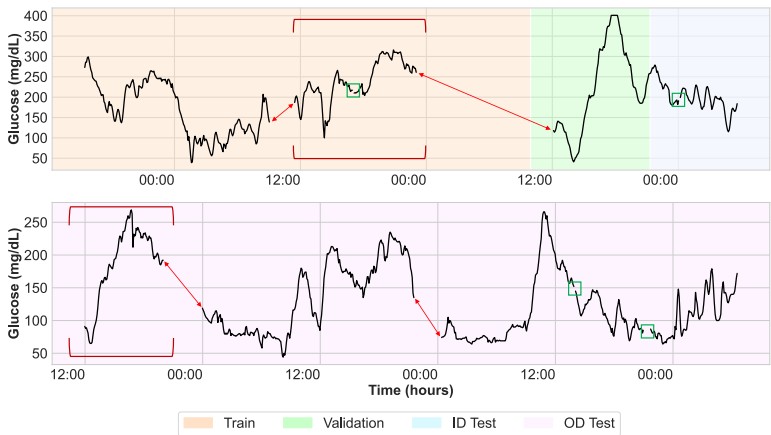

Figure 3: Example data processing on Weinstock et al. (2016). The red arrows denote segmentation, green blocks interpolate values, and red brackets indicate dropped segments due to length.

**Scaling.** We use min-max scaling to standardize the measurement range for both glucose values and covariates. The minimum and maximum values are computed per dataset using $\mathcal{D}_{tr}$, and then the same values are used to rescale $\mathcal{D}_{val}$, $\mathcal{D}_{id}$, and $\mathcal{D}_{od}$.

**Input-output pairs.** Let $\mathbf{x}^{(i)}_{j,k:k+L}$ be a length $L$ contiguous slice of a segment from index $k$ to $k + L$. We define an input-output pair as $\{\mathbf{x}^{(i)}_{j,k:k+L}, \mathbf{y}^{(i)}_{j,k+L+1:k+L+T}\}$, where $\mathbf{y}^{(i)}_{j,k+L+1:k+L+T} = \mathbf{x}^{(i)}_{j,k+L+1:k+L+T}$ and $T$ is the length of prediction interval. Our choices of $T$, $L$ and $k$ are as follows. In line with the previous works (Oviedo et al., 2017) we focus on the 1-hour ahead forecasting ($T = 12$ for 5 minute frequency). We treat $L$ as a hyper-parameter for model optimization since different models have different capabilities in capturing long-term dependencies. We sample $k$ without replacement from among the index set of the segment during training, similar to Oreshkin et al. (2020); Challu et al. (2023), and treat the total number of samples as a model hyper-parameter. We found the sampling approach to be preferable over the use of a sliding window with a unit stride (Herzen et al., 2022), as the latter is computationally prohibitive on larger training datasets and leads to high between-sample correlation, slowing convergence in optimization. We still use the sliding window when evaluating the model on the test set.

## 4 BENCHMARKS

### 4.1 TASKS AND METRICS

**Task 1: Predictive Accuracy.** Given the model prediction $\hat{\mathbf{y}}_{j,k+L:k+L+T}$, we measure accuracy on the test set using root mean squared error (RMSE) and mean absolute error (MAE):

$$RMSE_{i,j,k} = \sqrt{\frac{1}{T} \sum_{t=1}^{T} \left(y^{(i)}_{j,k+L+t} - \hat{y}^{(i)}_{j,k+L+t}\right)^2}; \quad MAE_{i,j,k} = \frac{1}{T} \sum_{t=1}^{T} \left|y^{(i)}_{j,k+L+t} - \hat{y}^{(i)}_{j,k+L+t}\right|.$$

Since the distribution of MAE and RMSE across samples is right-skewed, we use the median of the errors as has been done in Sergazinov et al. (2023); Armandpour et al. (2021).

**Task 2: Uncertainty Quantification.** To measure the quality of uncertainty quantification, we use two metrics: log-likelihood on test data and calibration. For models that estimate a parametric predictive distribution over the future values, $\hat{P}^{(i)}_{j,k+L+1:k+L+T} : \mathbb{R}^T \to [0, 1]$, we evaluate log-likelihood as

$$\log L_{i,j,k} = \log \hat{P}^{(i)}_{j,k+L+1:k+L+T}\left(\mathbf{y}^{(i)}_{j,k+L+1:k+L+T}\right),$$

where the parameters of the distribution are learned from training data, and the likelihood is evaluated on test data. Higher values indicate a better fit to the observed distribution. For both parametric

and non-parametric models (such as quantile-based methods), we use regression calibration metric (Kuleshov et al., 2018). The original metric is limited only to the univariate distributions. To address the issue, we report an average calibration across marginal estimates for each time $t = 1, \ldots, T$. To compute marginal calibration at time $t$, we (1) pick $M$ target confidence levels $0 < p_1 < \cdots < p_M < 1$; (2) estimate realized confidence level $\hat{p}_m$ using $N$ test input-output pairs as

$$\hat{p}_m = \frac{\left| \left\{ y_{j,k+L+t}^{(i)} | \hat{F}_{j,k+L+t}^{(i)} \left( y_{j,k+L+t}^{(i)} \right) \leq p_m \right\} \right|}{N};$$

and (3) compute calibration across all $M$ levels as

$$Cal_t = \sum_1^M (p_m - \hat{p}_m)^2.$$

The smaller the calibration value, the better the match between the estimated and true levels.

## 4.2 MODELS

To benchmark the performance on the two tasks, we compare the following models. **ARIMA** is a classical time-series model, which has been previously used for glucose predictions (Otoom et al., 2015; Yang et al., 2019). **Linear regression** is a simple baseline with a separate model for each time step $t = 1, \ldots, T$. **XGBoost** (Chen & Guestrin, 2016) is gradient-boosted tree method, with a separate model for each time step $t$ to support multi-output regression. **Transformer** represents a standard encoder-decoder auto-regressive Transformer implementation (Vaswani et al., 2017). **Temporal Fusion Transformer (TFT)** is a quantile-based model that uses RNN with attention. TFT is the only model that offers out-of-the-box support for static, dynamic known, and dynamic unknown covariates. **NHiTS** uses neural hierarchical interpolation for time series, focusing on the frequency domain (Challu et al., 2023). **Latent ODE** uses a recurrent neural network (RNN) to encode the sequence to a latent representation (Rubanova et al., 2019). The dynamics in the latent space are captured with another RNN with hidden state transitions modeled as an ODE. Finally, a generative model maps the latent space back to the original space. **Gluformer** is a probabilistic Transformer model that models forecasts using a mixture distribution (Sergazinov et al., 2023). For ARIMA, we use the code from (Federico Garza, 2022) which implements the algorithm from (Hyndman & Khandakar, 2008). For linear regression, XGBoost, TFT, and NHiTS, we use the open-source DARTS library (Herzen et al., 2022). For Latent ODE and Gluformer, we use the implementation in PyTorch (Rubanova et al., 2019; Sergazinov et al., 2023). We report the compute resources in Appendix C.

## 4.3 TESTING PROTOCOLS

In devising the experiments, we pursue the principles of reproducibility and fairness to all methods.

**Reproducibility.** As the performance results are data split dependent, we train and evaluate each model using the same two random splits. Additionally, all stochastically-trained models (tree-based and deep learning) are initialized 10 times on each training set with different random seeds. Thus, each stochastically-trained model is re-trained and re-evaluated 20 times, and each deterministically-trained model 2 times, with the final performance score taken as an average across evaluations. We report standard error of each metric across the re-runs in Appendix B.

**Fairness.** To promote fairness and limit the scope of comparisons, we focus on out-of-the-box model performance when establishing baselines. Thus, we do not consider additional model-specific tuning that could lead to performance improvements, e.g., pre-training, additional loss functions, data augmentation, distillation, learning rate warm-up, learning rate decay, etc. However, since model hyper-parameters can significantly affect performance, we automate the selection of these parameters. For ARIMA, we use the native automatic hyper-parameter selection algorithm provided in (Hyndman & Khandakar, 2008). For all other models, we use Optuna (Akiba et al., 2019) to run Bayesian optimization with a fixed budget of 50 iterations. We provide a discussion on the selected optimal model hyperparameters for each dataset in the supplement (Appendix C).

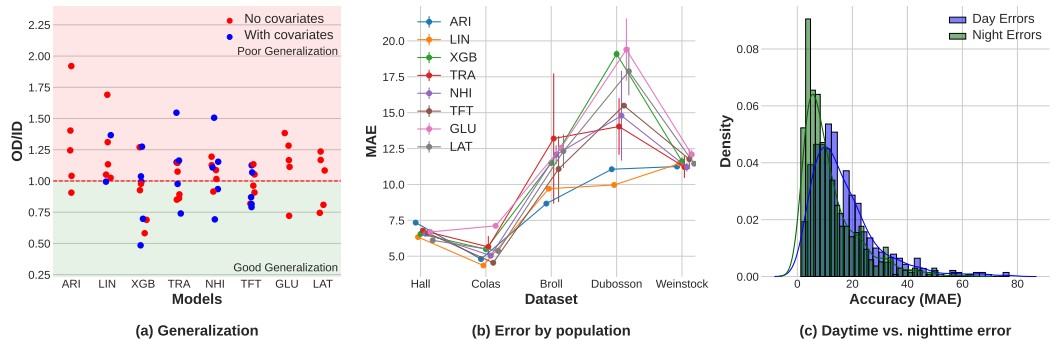

Figure 4: Analysis of errors by: (a) OD versus ID, (b) population diabetic type (healthy → Type 2 → Type 1), (c) daytime (9:00AM to 9:00PM) versus nighttime (9:00PM to 9:00AM).

Table 4: Accuracy and uncertainty metrics for selected models based on in-distribution (ID) test set without covariates. The selected models are best on at least one dataset for at least one metric. The best results on each data set are highlighted in **boldface**. TFT lacks likelihood information as it is a quantile-based model. Standard errors are reported in Appendix B.

| Accuracy | Broll | | Colas | | Dubosson | | Hall | | Weinstock | |
|---|---|---|---|---|---|---|---|---|---|---|
| | RMSE | MAE | RMSE | MAE | RMSE | MAE | RMSE | MAE | RMSE | MAE |
| ARIMA | **10.53** | **8.67** | 5.80 | 4.80 | 13.53 | 11.06 | 8.63 | 7.34 | 13.40 | 11.25 |
| Linear | 11.68 | 9.71 | **5.26** | **4.35** | **12.07** | **9.97** | 7.38 | 6.33 | 13.60 | 11.46 |
| Latent ODE | 14.37 | 12.32 | 6.28 | 5.37 | 20.14 | 17.88 | **7.13** | **6.11** | 13.54 | 11.45 |
| Transformer | 15.12 | 13.20 | 6.47 | 5.65 | 16.62 | 14.04 | 7.89 | 6.78 | **13.22** | **11.22** |
| Uncertainty | Lik. | Cal. | Lik. | Cal. | Lik. | Cal. | Lik. | Cal. | Lik. | Cal. |
| Gluformer | **-2.11** | **0.05** | **-1.07** | 0.14 | **-2.15** | **0.06** | **-1.56** | **0.05** | **-2.50** | 0.08 |
| TFT | – | 0.16 | – | **0.07** | – | 0.23 | – | 0.07 | – | **0.07** |

## 4.4 RESULTS

We trained and tested each model outlined above on all five datasets using the established protocols. Table 4 present the results for the best-performing models on Task 1 (predictive accuracy) and Task 2 (uncertainty quantification). Appendix B includes full tables for all models together with standard errors and the visualized forecasts for the best models on (Weinstock et al., 2016) dataset.

On Task 1, the simple ARIMA and linear regression models have the highest accuracy on all but two datasets. On Hall et al. (2018) dataset (mixed subjects including normoglycemic, prediabetes and Type 2 diabetes), the Latent ODE model performs the best. On Weinstock et al. (2016) dataset (the largest dataset), the Transformer model performs the best.

On Task 2, Gluformer model achieves superior performance as measured by model likelihood on all datasets. In regards to calibration, Gluformer is best on all but two datasets. On Colás et al. (2019) and Weinstock et al. (2016) datasets (the largest datasets), the best calibration is achieved by TFT.

## 5 ANALYSIS

### 5.1 WHY DOES THE PERFORMANCE OF THE MODELS DIFFER BETWEEN THE DATASETS?

Three factors consistently impact the results across the datasets and model configurations: (1) dataset size, (2) patients' composition, and (3) time of day. Below we discuss the effects of these factors on accuracy (Task 1), similar observations hold for uncertainty quantification (Task 2).

Tables 4 indicates that the best-performing model on each dataset is dependent on the dataset size. For smaller datasets, such as Broll et al. (2021) and Dubosson et al. (2018), simple models like ARIMA and linear regression yield the best results. In general, we see that deep learning models excel on

Table 5: Change in accuracy and uncertainty tasks between ID and OD sets. We indicate increases in performance in blue and decreases in red. TFT lacks likelihood information as it is a quantile-based model.

| Accuracy | Broll | | Colas | | Dubosson | | Hall | | Weinstock | |
|---|---|---|---|---|---|---|---|---|---|---|
| | RMSE | MAE | RMSE | MAE | RMSE | MAE | RMSE | MAE | RMSE | MAE |
| ARIMA | +11.59% | +12.01% | +2.02% | +1.49% | +38.56% | +31.81% | -4.79% | -5.08% | +18.47% | +18.56% |
| Linear | +2.51% | +1.29% | +1.27% | +1.38% | +30.01% | +19.41% | +6.46% | +4.58% | +14.5% | +14.22% |
| Latent ODE | +4.12% | +5.93% | -10.05% | -9.83% | -13.7% | -15.47% | +8.07% | +8.18% | +11.17% | +11.08% |
| Transformer | -7.16% | -6.96% | -7.78% | -7.23% | -5.52% | -7.52% | +3.69% | +4.15% | +7.0% | +6.16% |
| Uncertainty | Lik. | Cal. | Lik. | Cal. | Lik. | Cal. | Lik. | Cal. | Lik. | Cal. |
| Gluformer | +6.72% | +106.76% | -50.33% | -29.83% | +45.64% | +83.63% | +7.69% | +9.24% | +3.33% | +6.23% |
| TFT | – | -4.8% | – | +15.18% | – | +10.56% | – | +30.52% | – | +9.94% |

Table 6: Changes in accuracy and uncertainty tasks with and without covariates on ID test set. We indicate increases in performance in blue and decreases in red. TFT lacks likelihood information as it is a quantile-based model.

| Accuracy | Broll | | Colas | | Dubosson | | Hall | | Weinstock | |
|---|---|---|---|---|---|---|---|---|---|---|
| | RMSE | MAE | RMSE | MAE | RMSE | MAE | RMSE | MAE | RMSE | MAE |
| Linear | -14.82% | -13.34% | +5.54% | +5.75% | +2.84% | +0.61% | +6.17% | +5.09% | -1.54% | -1.08% |
| Transformer | -15.14% | -14.64% | +30.31% | +37.56% | +64.99% | +73.82% | -5.06% | -5.4% | +9.33% | +12.41% |
| Uncertainty | Lik. | Cal. | Lik. | Cal. | Lik. | Cal. | Lik. | Cal. | Lik. | Cal. |
| TFT | – | +94.6% | – | +114.61% | – | +7.57% | – | +16.84% | – | -21.55% |

larger datasets: Hall et al. (2018) (best model is Latent ODE) and Weinstock et al. (2016) (best model is Transformer) are 2 of the largest datasets. The only exception is Colás et al. (2019), on which the best model is linear regression. We suggest that this could be explained by the fact that despite being large, Colás et al. (2019) dataset has low number observations per patient: 100,000 glucose readings across 200 patients yeilds 500 readings or 2 days worth of data per patient. In comparison, Hall et al. (2018) has 2,000 readings per patient or 7 days, and Weinstock et al. (2016) has approximately 3,000 readings per patient or 10 days.

Figure 4(b) demonstrates that the accuracy of predictions is substantially influenced by the patients' population group. Healthy subjects demonstrate markedly smaller errors compared to subjects with Type 1 or Type 2 diabetes. This discrepancy is due to healthy subjects maintaining a narrower range of relatively low glucose level, simplifying forecasting. Patients with Type 1 exhibit larger fluctuations partly due to consistently required insulin administration in addition to lifestyle-related factors, whereas most patients with Type 2 are not on insulin therapy.

Figure 4(c) shows the impact of time of day on accuracy, with daytime (defined as 9:00AM to 9:00PM) being compared to nighttime for Transformer model on Broll et al. (2021) dataset. The distribution of daytime errors is more right-skewed and right-shifted compared to the distribution of nighttime errors, signifying that daytime glucose values are harder to predict. Intuitively, glucose values are less variable overnight due to the absence of food intake and exercise, simplifying forecasting. We include similar plots for all models and datasets in Appendix B. This finding underscores the importance of accounting for daytime and nighttime when partitioning CGM data for model training and evaluation.

Overall, we recommend using simpler shallow models when data is limited, the population group exhibits less complex CGM profiles (such as healthy individuals or Type 2 patients), or for nighttime forecasting. Conversely, when dealing with larger and more complex datasets, deep or hybrid models are the preferred choice. In clinical scenarios where data is actively collected, it is advisable to deploy simpler models during the initial stages and, in later stages, maintain an ensemble of both shallow and deep models. The former can act as a guardrail or be used for nighttime predictions.

## 5.2 ARE THE MODELS GENERALIZABLE TO PATIENTS BEYOND THE TRAINING DATASET?

Table 5 compares accuracy and uncertainty quantification of selected models on in-distribution (ID) and out-of-distribution (OD) test sets, while the full table is provided in Appendix B. Here we assume that each patient is different, in that the OD set represents a distinct distribution from the ID set.

In both tasks, most models exhibit decreased performance on the OD data, emphasizing individual-level variation between patients and the difficulty of cold starts on new patient populations. Figure 4(a) displays OD-to-ID accuracy ratio (measured in MAE) for each model and dataset: higher ratios indicate poorer generalization, while lower ratios indicate better generalization. In general, we observe that deep learning models (Transformer, NHiTS, TFT, Gluformer, and Latent ODE) generalize considerably better than the simple baselines (ARIMA and linear regression). We attribute this to the deep learning models' ability to capture and recall more patterns from the data. Notably, XGBoost also demonstrates strong generalization capabilities and, in some instances, outperforms the deep learning models in the generalization power.

### 5.3 How does adding the covariates affect the modeling quality?

Table 6 demonstrates the impact of including covariates in the models on Task 1 (accuracy) and Task 2 (uncertainty quantification) compared to the same models with no covariates. As the inclusion of covariates represents providing model with more information, any changes in performance can be attributed to (1) the quality of the covariate data; (2) model's ability to handle multiple covariates. We omit ARIMA, Gluformer, and Latent ODE models as their implementations do not support covariates.

In both tasks, the impact of covariates on model performance varies depending on the dataset. For Colás et al. (2019) and Dubosson et al. (2018), we observe a decrease in both accuracy and uncertainty quantification performance with the addition of covariates. Given that these are smaller datasets with a limited number of observations per patient, we suggest that the inclusion of covariates leads to model overfitting, consequently increasing test-time errors. In contrast, for Broll et al. (2021) that is also small, unlike for all other datasets, we have covariates extracted solely from the timestamp, which appears to enhance model accuracy. This increase in performance is likely attributable to all patients within the train split exhibiting more pronounced cyclical CGM patterns, which could explain why the overfitted model performs better. This is further supported by the fact that the performance on the OD set deteriorates with the addition of covariates. Finally, in the case of Hall et al. (2018) and Weinstock et al. (2016), which are large datasets, the inclusion of covariates has mixed effects, indicating that covariates do not contribute significantly to the model's performance.

## 6 Discussion

**Impact.** We discuss potential negative societal impact of our work. **First**, inaccurate glucose forecasting could lead to severe consequences for patients. This is by far the most important consideration that we discuss further in Appendix D. **Second**, there is a potential threat from CGM device hacking that could affect model predictions. **Third**, the existence of pre-defined tasks and datasets may stifle research, as researchers might focus on overfitting and marginally improving upon well-known datasets and tasks. **Finally**, the release of health records must be treated with caution to guarantee patients' right to privacy.

**Future directions.** We outline several research avenues: (1) adding new public CGM datasets and tasks; (2) open-sourcing physiological and hybrid models; (3) exploring model training augmentation, such as pre-training on aggregated data followed by patient-specific fine-tuning and down-sampling night periods; (4) developing scaling laws for dataset size and model performance; and (5) examining covariate quality and principled integration within models. Related to the point (5), we note that out of the 5 collected datasets, only Dubosson et al. (2018) records time-varying covariates describing patients physical activity (e.g. accelerometer readings, heart rate), blood pressure, food intake, and medication. We believe having larger datasets that comprehensively track dynamic patient behavior could lead to new insights and more accurate forecasting.

## 7 Conclusion

In this work, we have presented a comprehensive resource to address the challenges in CGM-based glucose trajectory prediction, including a curated repository of public datasets, a standardized task list, a set of benchmark models, and a detailed analysis of performance-influencing factors. Our analysis emphasizes the significance of dataset size, patient population, testing splits (e.g., in- and out-of-distribution, daytime, nighttime), and covariate availability.

ACKNOWLEDGEMENTS

The source of a subset of the data is the T1D Exchange, but the analyses, content, and conclusions presented herein are solely the responsibility of the authors and have not been reviewed or approved by the T1D Exchange.

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

# A    DATASETS

**Previous works.** We summarize previous work on the CGM datasets in Table 7.

Table 7: Summary of the glucose prediction models by dataset and model type. We indicate "open" for datasets that are publicly available online, and "proprietary" for the ones that cannot be released.

| Dataset | Diabetes | # | Deep | Shallow | Physiological |
|---|---|---|---|---|---|
| DirecNet, 2008 | Type 1 | 30 | He & Wang (2020) | Eren-Oruklu et al. (2010) | Balakrishnan et al. (2013); Chen & Tsai (2010) |
| Anthimopoulos et al. (2015) | Type 1 | 20 | Sun et al. (2018) | | |
| Mauras et al. (2012) | Type 1 | 146 | Indrawan et al. (2021) | | |
| Georga et al. (2009) | Type 1 | 15 | | Georga et al. (2015) | |
| Marling & Bunescu (2020) | Type 1 | 12 | Deng et al. (2021); van Doorn et al. (2021); Zhu et al. (2022a); Martinsson et al. (2020) | van Doorn et al. (2021) | |
| Dubosson et al. (2018) | Type 1 | 9 | Munoz-Organero (2020) | | |
| Aleppo et al. (2017) | Type 1 | 168 | Jaloli & Cescon (2022) | | |
| Fox et al. (2018) | Type 1 | 40 | Fox et al. (2018); Armandpour et al. (2021); Sergazinov et al. (2023) | | |
| Cescon (2013) | Type 1 | 59 | Jaloli & Cescon (2022) | | |
| **Total (open)** | | | **13** | **3** | **2** |
| Simulation | NA | NA | Li et al. (2020); Langarica et al. (2023); Liu et al. (2018) | Reymann et al. (2016); Boiroux et al. (2012); Otoom et al. (2015) | Boiroux et al. (2012); Bock et al. (2015); Calm et al. (2011); De Pereda et al. (2012); Fang et al. (2015); Laguna et al. (2014b) |
| Proprietary | NA | 1-851 | Xu et al. (2022); Li et al. (2020); Prendin et al. (2021); Aliberti et al. (2019); Liu et al. (2018); Ben Ali et al. (2018); Shi et al. (2015) | Prendin et al. (2021); Yang et al. (2019); Sudharsan et al. (2015); Hidalgo et al. (2017); Efendic et al. (2014); Botwey et al. (2014); Wang et al. (2013); Zarkogianni et al. (2014) | Gyuk et al. (2019); Novara et al. (2016); Bock et al. (2015); Duun-Henriksen et al. (2013); Laguna et al. (2014a); Wu et al. (2011); Zhang et al. (2016) |
| **Total (proprietary)** | | | **10** | **11** | **13** |

**Access.** The datasets are distributed according to the following licences and can be downloaded from the following links:

1. Broll et al. (2021)      License: GPL-2                    Source: link
2. Colás et al. (2019)      License: Creative Commons 4.0    Source: link
3. Dubosson et al. (2018)   License: Creative Commons 4.0    Source: link
4. Hall et al. (2018)       License: Creative Commons 4.0    Source: link
5. Weinstock et al. (2016)  License: Creative Commons 4.0    Source: link

**Covariates.** We summarize covariate types for each dataset in Table 8. For each dataset, we extract the following dynamic known covariates from the time stamp: year, month, day, hour, minute, and second (only for Broll). Broll provides no covariates aside from the ones extracted from the time stamp. Colas, Hall, and Weinstock provide demographic information for the patients (static covariates). Dubosson is the only dataset for which dynamic unknown covariates such as heart rate, insulin levels, and blood pressure are available.

Table 8: Covariate information for each dataset.

| | Covariate | Broll | Colas | Dubosson | Hall | Weinstock |
|---|---|---|---|---|---|---|
| Static | Age | | ✓ | | ✓ | |
| | Height | | | | ✓ | ✓ |
| | ... | | | | | |
| | **Total** | 0 | 7 | 0 | 48 | 38 |
| Dyn. Kn. | Year | ✓ | ✓ | ✓ | ✓ | ✓ |
| | Month | ✓ | ✓ | ✓ | ✓ | ✓ |
| | ... | | | | | |
| | **Total** | 6 | 5 | 5 | 5 | 5 |
| Dyn. Unkn. | Insulin | | | ✓ | | |
| | Heart Rate | | | ✓ | | |
| | ... | | | | | |
| | **Total** | 0 | 0 | 11 | 0 | 0 |

# B ANALYSIS

## B.1 VISUALIZED PREDICTIONS

We provide visualized forecasts for the same 5 segments of Weinstock et al. (2016) data for the best performing models: linear regression, Latent ODE (Rubanova et al., 2019), and Transformer (Vaswani et al., 2017) on Task 1 (accuracy), and Gluformer (Sergazinov et al., 2023) and TFT (Lim et al., 2021) on Task 2 (uncertainty). For the best models on Task 2, we also provide the estimated confidence intervals or the predictive distribution, whichever is available. For visualization, we have truncated the input sequence to 1 hour (12 points); however, we note that different models have different input length and require at least 4 hours of observations to forecast the future trajectory.

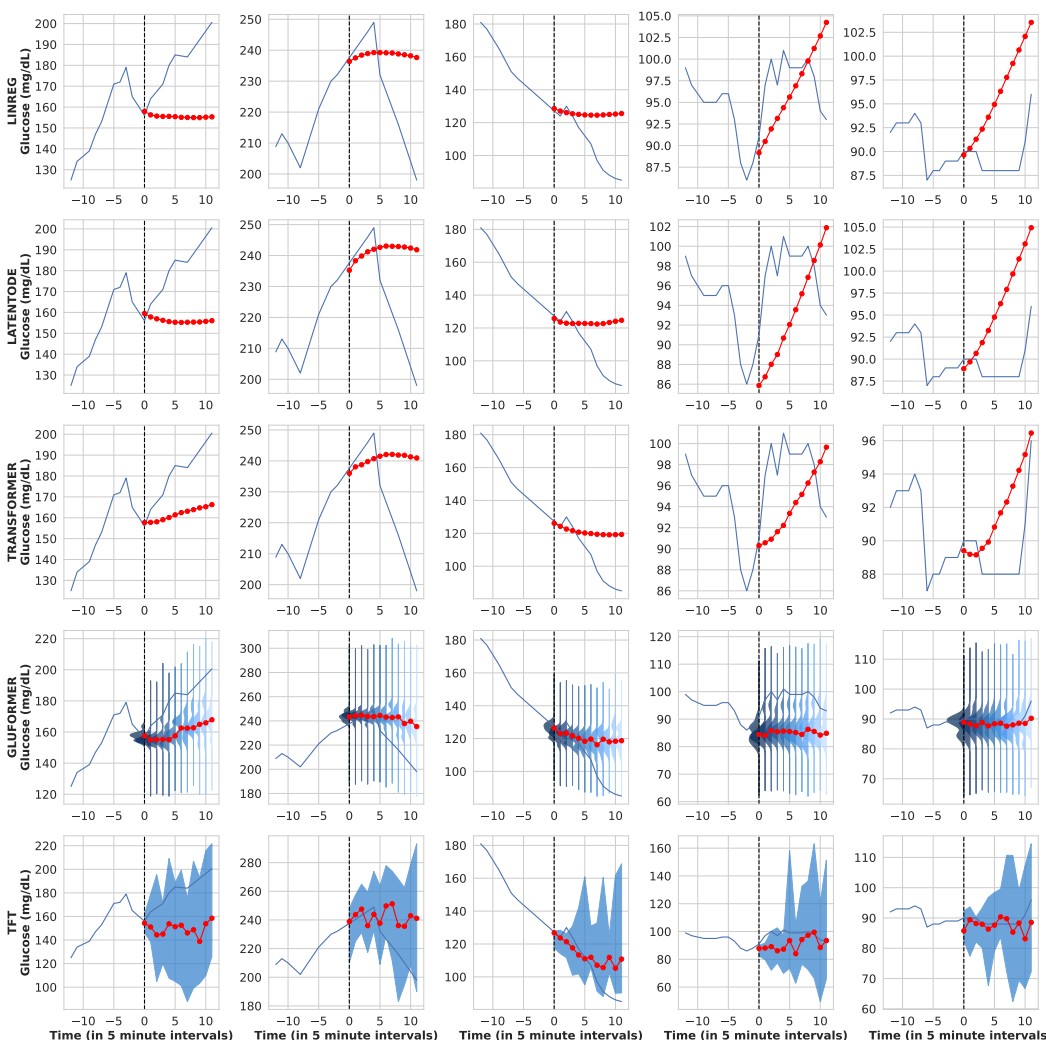

Figure 5: Model forecasts on Weinstock et al. (2016) dataset.

## B.2 PERFORMANCE

For reference, we include results for all models both with and without covariates evaluated on ID and OD splits in Table 9 on Task 1 (accuracy) and in Table 10 on Task 2 (uncertainty quantification).

Table 9: Model results on the data sets for Task 1 (accuracy).

| | $p(\cdot|x)$ | Data | Broll RMSE | Broll MAE | Colas RMSE | Colas MAE | Dubosson RMSE | Dubosson MAE | Hall RMSE | Hall MAE | Weinstock RMSE | Weinstock MAE |
|---|---|---|---|---|---|---|---|---|---|---|---|---|
| ARI | ✗ | ID | 10.53 | 8.67 | 5.80 | 4.80 | 13.53 | 11.06 | 8.63 | 7.34 | 13.40 | 11.25 |
| ARI | ✗ | OD | 11.75 | 9.71 | 5.91 | 4.87 | 18.75 | 14.58 | 8.22 | 6.97 | 15.87 | 13.34 |
| min Δ(ID, OD)% | | | +11.8% | | +1.75% | | +35.18% | | -4.94% | | +18.51% | |
| LIN | ✗ | ID | 11.68 | 9.71 | 5.26 | 4.35 | 12.07 | 9.97 | 7.38 | 6.33 | 13.60 | 11.46 |
| LIN | ✓ | ID | 9.95 | 8.41 | 5.56 | 4.60 | 12.41 | 10.03 | 7.84 | 6.66 | 13.39 | 11.34 |
| Improv. | | | -14.08% | | +5.65% | | +1.73% | | +5.63% | | -1.31% | |
| LIN | ✗ | OD | 11.98 | 9.83 | 5.33 | 4.41 | 15.69 | 11.90 | 7.86 | 6.62 | 15.58 | 13.09 |
| LIN | ✓ | OD | 23.30 | 16.80 | 5.54 | 4.57 | 203114.47 | 67548.59 | 14.22 | 10.02 | 15.66 | 13.16 |
| Improv. | | | +82.7% | | +3.8% | | +930929.99% | | +66.07% | | +0.54% | |
| min Δ(ID, OD)% | | | +1.9% | | -0.45% | | +24.71% | | +5.52% | | +14.36% | |
| XGB | ✗ | ID | 12.80 | 11.50 | 6.42 | 5.49 | 21.18 | 19.09 | 7.58 | 6.55 | 13.63 | 11.61 |
| XGB | ✓ | ID | 13.89 | 11.87 | 6.37 | 5.46 | 20.89 | 18.55 | 8.05 | 7.02 | 13.77 | 11.77 |
| Improv. | | | +5.88% | | -0.61% | | -2.08% | | +6.67% | | +1.18% | |
| XGB | ✗ | OD | 9.76 | 8.72 | 6.18 | 5.32 | 17.57 | 15.42 | 7.49 | 6.52 | 15.36 | 13.04 |
| XGB | ✓ | OD | 9.67 | 8.56 | 6.36 | 5.47 | 17.44 | 15.46 | 8.20 | 7.11 | 15.55 | 13.43 |
| Improv. | | | -1.37% | | +2.91% | | -0.26% | | +9.25% | | +2.09% | |
| min Δ(ID, OD)% | | | -29.14% | | -3.47% | | -18.11% | | -0.84% | | +12.51% | |
| GLU | ✗ | ID | 14.19 | 12.55 | 8.17 | 7.12 | 21.74 | 19.40 | 7.74 | 6.69 | 14.07 | 12.09 |
| GLU | ✗ | OOD | 16.70 | 14.82 | 6.94 | 6.03 | 23.48 | 20.70 | 8.17 | 7.04 | 15.94 | 13.65 |
| min Δ(ID, OD)% | | | +17.85% | | -15.17% | | +7.37% | | +5.39% | | +13.08% | |
| LAT | ✗ | ID | 14.37 | 12.32 | 6.28 | 5.37 | 20.14 | 17.88 | 7.13 | 6.11 | 13.54 | 11.45 |
| LAT | ✗ | OOD | 14.96 | 13.05 | 5.64 | 4.84 | 17.38 | 15.12 | 7.71 | 6.61 | 15.06 | 12.72 |
| min Δ(ID, OD)% | | | +5.03% | | -9.94% | | -14.59% | | +8.12% | | +11.12% | |
| NHI | ✗ | ID | 13.79 | 12.07 | 5.93 | 5.04 | 17.45 | 14.79 | 7.68 | 6.57 | 13.29 | 11.21 |
| NHI | ✓ | ID | 16.20 | 14.64 | 9.09 | 8.03 | 30.43 | 27.97 | 8.16 | 7.10 | 13.41 | 11.31 |
| Improv. | | | +19.36% | | +56.31% | | +81.74% | | +7.13% | | +0.9% | |
| NHI | ✗ | OD | 14.64 | 12.77 | 5.68 | 4.83 | 18.20 | 15.59 | 7.74 | 6.62 | 14.52 | 12.24 |
| NHI | ✓ | OD | 15.66 | 14.01 | 7.56 | 6.65 | 37.35 | 33.52 | 8.59 | 7.53 | 14.40 | 12.12 |
| Improv. | | | +8.35% | | +35.49% | | +110.09% | | +12.34% | | -0.91% | |
| min Δ(ID, OD)% | | | -3.8% | | -17.01% | | +4.86% | | +0.78% | | +7.29% | |
| TFT | ✗ | ID | 13.73 | 11.07 | 5.62 | 4.54 | 18.37 | 15.49 | 7.92 | 6.61 | 14.32 | 11.76 |
| TFT | ✓ | ID | 14.68 | 12.43 | 6.51 | 5.27 | 18.43 | 15.51 | 8.42 | 7.06 | 14.97 | 12.30 |
| Improv. | | | +9.53% | | +15.9% | | +0.22% | | +6.52% | | +4.55% | |
| TFT | ✗ | OD | 12.43 | 10.23 | 5.51 | 4.47 | 17.50 | 14.53 | 8.12 | 6.76 | 15.25 | 12.50 |
| TFT | ✓ | OD | 13.25 | 11.17 | 5.79 | 4.68 | 17.19 | 14.43 | 8.93 | 7.44 | 15.47 | 12.67 |
| Improv. | | | +7.91% | | +4.84% | | -1.22% | | +10.01% | | +1.41% | |
| min Δ(ID, OD)% | | | -9.91% | | -11.11% | | -6.84% | | +2.41% | | +3.18% | |
| TRA | ✗ | ID | 15.12 | 13.20 | 6.47 | 5.65 | 16.62 | 14.04 | 7.89 | 6.78 | 13.22 | 11.22 |
| TRA | ✓ | ID | 12.83 | 11.27 | 8.44 | 7.77 | 27.43 | 24.40 | 7.49 | 6.42 | 14.46 | 12.61 |
| Improv. | | | -14.89% | | +33.93% | | +69.41% | | -5.23% | | +10.87% | |
| TRA | ✗ | OD | 14.04 | 12.28 | 5.97 | 5.24 | 15.71 | 12.98 | 8.18 | 7.07 | 14.15 | 11.91 |
| TRA | ✓ | OD | 13.76 | 12.13 | 7.26 | 6.59 | 34.11 | 28.21 | 7.40 | 6.29 | 15.59 | 13.58 |
| Improv. | | | -1.59% | | +23.61% | | +117.27% | | -10.23% | | +12.14% | |
| min Δ(ID, OD)% | | | -7.06% | | -14.62% | | -6.52% | | -1.57% | | +6.58% | |

## B.3 FEATURE IMPORTANCE

Based on the performance results reported in Tables 9 on Task 1 (accuracy) and in Table 10, XGBoost (Chen & Guestrin, 2016) is the only model that consistently works better with inclusion of extraneous covariates, improves in accuracy on 3 out of 5 datasets and uncertainty quantification on 4 out of 5 datasets. Table 11 lists the top 10 covariates selected by XGBoost for each dataset. For time-varying

Table 10: Model results on the data sets for Task 2 (uncertainty quantification).

| | $p(\cdot|x)$ | Data | Broll Lik.↑ | Broll Cal.↓ | Colas Lik.↑ | Colas Cal.↓ | Dubosson Lik.↑ | Dubosson Cal.↓ | Hall Lik.↑ | Hall Cal.↓ | Weinstock Lik.↑ | Weinstock Cal.↓ |
|---|---|---|---|---|---|---|---|---|---|---|---|---|
| ARI | ✗ | ID | -9.93 | 0.11 | -9.30 | 0.10 | -10.47 | 0.10 | -9.81 | 0.10 | -10.21 | 0.12 |
| | ✗ | OOD | -10.06 | 0.07 | -9.38 | 0.08 | -10.44 | 0.08 | -9.66 | 0.07 | -10.32 | 0.12 |
| min Δ(ID, OD)% | | | -1.30% | -36.06% | -0.86% | -19.95% | +0.28% | -20.07% | +1.52% | -29.92% | -1.07% | +0.06% |
| LIN | ✗ | ID | -9.89 | 0.12 | -9.19 | 0.15 | -10.10 | 0.18 | -9.56 | 0.10 | -10.14 | 0.11 |
| | ✓ | ID | -9.87 | 0.13 | -9.17 | 0.19 | -10.15 | 0.21 | -10.30 | 0.19 | -10.12 | 0.11 |
| Improv. | | | +0.15% | +5.99% | +0.25% | +24.47% | -0.41% | +11.39% | -7.67% | +97.36% | +0.13% | -1.67% |
| | ✗ | OD | -9.95 | 0.15 | -9.16 | 0.15 | -10.11 | 0.17 | -9.53 | 0.10 | -10.22 | 0.11 |
| | ✓ | OD | -10.24 | 0.55 | -9.16 | 0.17 | -12.08 | 0.48 | -10.42 | 0.23 | -11.13 | 0.21 |
| Improv. | | | -2.88% | +256.79% | +0.01% | +10.19% | -19.52% | +181.06% | -9.26% | +130.96% | -8.92% | +87.0% |
| min Δ(ID, OD)% | | | -0.65% | +24.98% | +0.33% | -8.88% | -0.02% | -7.62% | +0.33% | +4.43% | -0.85% | -3.1% |
| XGB | ✗ | ID | -9.94 | 0.07 | -9.42 | 0.10 | -10.55 | 0.07 | -9.68 | 0.09 | -10.20 | 0.11 |
| | ✓ | ID | -10.06 | 0.07 | -9.40 | 0.09 | -10.54 | 0.06 | -9.70 | 0.09 | -10.21 | 0.10 |
| Improv. | | | -1.22% | +0.59% | +0.12% | -7.2% | +0.13% | -6.98% | -0.31% | -1.3% | -0.15% | -4.62% |
| | ✗ | OD | -10.03 | 0.11 | -9.36 | 0.09 | -10.22 | 0.07 | -9.56 | 0.08 | -10.28 | 0.11 |
| | ✓ | OD | -10.03 | 0.11 | -9.38 | 0.08 | -10.20 | 0.07 | -9.53 | 0.10 | -10.31 | 0.10 |
| Improv. | | | -0.04% | +1.75% | -0.2% | -7.0% | +0.13% | -1.62% | +0.31% | +21.93% | -0.34% | -4.89% |
| min Δ(ID, OD)% | | | +0.29% | +67.42% | +0.59% | -4.55% | +3.17% | +5.02% | +1.77% | -14.37% | -0.83% | +2.16% |
| GLU | ✗ | ID | -2.11 | 0.05 | -1.07 | 0.14 | -2.15 | 0.06 | -1.56 | 0.05 | -2.50 | 0.08 |
| | ✗ | OOD | -1.96 | 0.11 | -1.61 | 0.10 | -1.17 | 0.12 | -1.44 | 0.06 | -2.41 | 0.09 |
| min Δ(ID, OD)% | | | +6.72% | +106.76% | -50.33% | -29.83% | +45.64% | +83.63% | +7.69% | +9.24% | +3.33% | +6.23% |
| LAT | ✗ | ID | -25.29 | 0.36 | -10.47 | 0.25 | -52.18 | 0.42 | -20.24 | 0.30 | -26.15 | 0.33 |
| | ✗ | OOD | -28.75 | 0.38 | -8.80 | 0.24 | -30.19 | 0.44 | -18.19 | 0.36 | -30.08 | 0.40 |
| min Δ(ID, OD)% | | | -13.67% | +6.5% | +15.89% | -4.01% | +42.14% | +4.59% | +10.12% | +20.58% | -15.03% | +20.72% |
| NHI | ✗ | ID | -10.01 | 0.12 | -9.32 | 0.11 | -10.37 | 0.10 | -9.62 | 0.09 | -10.13 | 0.11 |
| | ✓ | ID | -10.37 | 0.07 | -9.48 | 0.21 | -10.80 | 0.08 | -9.63 | 0.07 | -10.13 | 0.11 |
| Improv. | | | -3.63% | -37.79% | -1.68% | +91.12% | -4.19% | -21.68% | -0.07% | -24.77% | -0.01% | -5.46% |
| | ✗ | OD | -10.08 | 0.10 | -9.26 | 0.11 | -10.18 | 0.12 | -9.49 | 0.08 | -10.20 | 0.12 |
| | ✓ | OD | -10.21 | 0.06 | -9.36 | 0.14 | -11.10 | 0.20 | -9.58 | 0.06 | -10.19 | 0.11 |
| Improv. | | | -1.3% | -34.64% | -1.17% | +24.57% | -9.0% | +66.46% | -0.94% | -14.44% | +0.14% | -8.1% |
| min Δ(ID, OD)% | | | +1.55% | -16.3% | +1.23% | -34.06% | +1.76% | +15.16% | +1.38% | -14.79% | -0.55% | +4.17% |
| TFT | ✗ | ID | – | 0.16 | – | 0.07 | – | 0.23 | – | 0.07 | – | 0.07 |
| | ✓ | ID | – | 0.30 | – | 0.16 | – | 0.25 | – | 0.08 | – | 0.06 |
| Improv. | | | –% | +94.6% | –% | +114.61% | –% | +7.57% | –% | +16.84% | –% | -21.55% |
| | ✗ | OD | – | 0.15 | – | 0.09 | – | 0.26 | – | 0.08 | – | 0.08 |
| | ✓ | OD | – | 0.23 | – | 0.09 | – | 0.35 | – | 0.08 | – | 0.05 |
| Improv. | | | –% | +57.74% | –% | +0.35% | –% | +37.5% | –% | -1.64% | –% | -35.43% |
| min Δ(ID, OD)% | | | –% | -22.83% | –% | -46.14% | –% | +10.56% | –% | +9.88% | –% | -9.51% |
| TRA | ✗ | ID | -9.99 | 0.23 | -9.37 | 0.21 | -10.36 | 0.12 | -9.60 | 0.13 | -10.12 | 0.11 |
| | ✓ | ID | -10.11 | 0.21 | -9.45 | 0.31 | -10.68 | 0.10 | -9.60 | 0.10 | -10.15 | 0.11 |
| Improv. | | | -1.21% | -6.84% | -0.79% | +45.69% | -3.05% | +48.29% | +0.05% | -27.4% | -0.34% | +0.16% |
| | ✗ | OD | -9.98 | 0.19 | -9.30 | 0.22 | -10.09 | 0.14 | -9.47 | 0.15 | -10.17 | 0.12 |
| | ✓ | OD | -10.02 | 0.11 | -9.36 | 0.22 | -10.63 | 0.25 | -9.49 | 0.08 | -10.20 | 0.12 |
| Improv. | | | -0.41% | -43.28% | -0.65% | +1.0% | -5.32% | +73.94% | -0.16% | -45.49% | -0.33% | +2.63% |
| min Δ(ID, OD)% | | | +0.93% | -47.95% | +0.92% | -28.49% | +2.59% | +17.77% | +1.34% | -14.35% | -0.53% | +8.99% |

features, such as the 36 heart rate observations in Dubosson, the maximum importance across the input length is considered as the feature importance. Below, we provide a discussion on the selected features.

Among features available for all datasets, dynamic time features, such as hour and day of the week, consistently appear in the top 3 important features across all datasets. This could serve as indication that patients tend to adhere to daily routines; therefore, including time features helps the model to predict more accurately. At the same time, patient unique identifier does not appear to be important, only appearing in the top 10 for Broll (Broll only has 7 covariates in total) and Colas. This could be indicative of the fact that differences between patients is explained well by other covariates.

Dynamic physical activity features such as heart rate and blood pressure are only available for Dubosson. Based on the table, we see that medication intake, heart rate and blood pressure metrics, and physical activity measurements are all selected by XGBoost as highly important.

Demographic and medical record information is not available for Broll and Dubosson. For the rest of the datatsets, we observe medication (e.g. Vitamin D, Lisinopril for Weinstock), disease indicators (e.g. Diabetes T2 for Colas, Osteoporosis for Weinstock), health summary metrics (Body Mass Index for Colas), as well as indices derived from CGM measurements (e.g. J-index (Wójcicki, 1995)) being selected as highly important.

Table 11: Top-10 features with importance weights selected by XGBoost for each dataset.

| Broll | | Colas | | Dubosson | | Hall | | Weinstock | |
|---|---|---|---|---|---|---|---|---|---|
| Covariate | Importance | Covariate | Importance | Covariate | Importance | Covariate | Importance | Covariate | Importance |
| Month | 0.001428 | Hour | 0.000634 | Slow Insulin Intake | 0.000625 | Day of week | 0.002322 | Minute | 0.000444 |
| Day of week | 0.001144 | Day of week | 0.000202 | Hour | 0.000390 | Median CGM | 0.002044 | Day of week | 0.000365 |
| Second | 0.000886 | Glycemia | 0.000133 | heart Rate Variability Index | 0.000359 | J Index of CGM | 0.001808 | Hour | 0.000291 |
| Hour | 0.000768 | Minute | 0.000126 | Body Temperature | 0.000323 | Hour | 0.001786 | Vitamin D | 0.000197 |
| Minute | 0.000410 | Diabetes T2 | 0.000121 | Posture | 0.000296 | Freq. High CGM | 0.001576 | Year | 0.000194 |
| Patient ID | 0.000072 | Patient ID | 0.000104 | Activity | 0.000292 | Freq. Low CGM | 0.001153 | Erectile dysfunction | 0.000154 |
| Year | 0.000000 | # Follow Up Visits | 0.000093 | Calories | 0.000291 | Minute | 0.001136 | Osteoporosis | 0.000140 |
| | | Body Mass Index (BMI) | 0.000090 | Heart Rate | 0.000197 | % Pre-Diabetec CGM | 0.001054 | Chronic kidney disease | 0.000140 |
| | | Age | 0.000085 | Blood Pressure | 0.000190 | Coefficient of CGM Variation | 0.000779 | # of Meter Checks per Day | 0.000137 |
| | | Gender | 0.000070 | Fast Insulin Intake | 0.000140 | Variance of CGM | 0.000706 | Lisinopril | 0.000128 |

## B.4 STABILITY

Reproducible model performance is crucial in the clinical settings. In Table 16, we report standard deviation of MAE across random data splits. As expected, the smallest datasets (Broll et al. (2021) and Dubosson et al. (2018)) have largest variability. The number of patients in Broll et al. (2021) and Dubosson et al. (2018) is 5 and 9, respectively, thus randomly selecting 1 subject for OD test set has large impact on the model performance as the training set is altered drastically.

Prior works on deep learning has found that initial weights can have large impact on the performance (LeCun et al., 2002; Sutskever et al., 2013; Zhu et al., 2022b). Therefore, we re-run each deep learning model 10 times with random initial weights for each data split and report the average. We also report standard deviation of deep learning model results across random model initializations (indicated in parentheses). We find that good initialization indeed matters as we observe that the results differ across re-runs with different starting weights. Such behavior could be undesirable in the clinical settings as the model training cannot be automated. The Transformer is the only robust deep learning model that consistently converges to the same results irrespective of the initial weights, which is reflected in near 0 standard deviation. At the same time, Transformer-based models such as Gluformer and TFT do not exhibit this feature.

We include standard errors of each metric: RMSE (Task 1) in Table 12, MAE (Task 1) in Table 16, likelihood (Task 2) in Table 14, and calibration in Table 15. For deep learning models, there are 2 sources of randomness: random data split and model initialization. Therefore, we report two values for standard deviation: one across data splits (averaged over model initializations) and one for model initialization (averaged across data splits).

## B.5 DAYTIME VERSUS NIGHTTIME ERROR DISTRIBUTION

We provide daytime and nighttime error (MAE) distribution for all models and datasets in Figure 6. In general, we note that for larger datasets (Colas and Weinstock), the difference in daytime and nighttime error distribution appears smaller.

Table 12: Standard error of MSE across data splits and model random initializations.

| | $p(\cdot|x)$ | Data | Broll | Colas | Dubosson | Hall | Weinstock |
|---|---|---|---|---|---|---|---|
| ARI | ✗ | ID | 110.90 +- 21.95 | 33.60 +- 0.68 | 183.11 +- 40.58 | 74.52 +- 2.25 | 179.54 +- 3.14 |
| ARI | ✗ | OD | 138.08 +- 52.73 | 34.97 +- 5.65 | 351.53 +- 227.93 | 67.55 +- 25.18 | 251.97 +- 18.59 |
| LIN | ✗ | ID | 136.49 +- 13.04 | 27.70 +- 0.33 | 145.65 +- 30.12 | 54.51 +- 1.67 | 185.04 +- 2.53 |
| LIN | ✓ | ID | 99.04 +- 7.45 | 30.86 +- 0.71 | 154.04 +- 32.62 | 61.45 +- 3.42 | 179.38 +- 2.56 |
| LIN | ✗ | OD | 143.43 +- 50.18 | 28.41 +- 2.55 | 246.19 +- 148.20 | 61.78 +- 17.90 | 242.59 +- 21.51 |
| LIN | ✓ | OD | 542.88 +- 617.76 | 30.69 +- 2.55 | 41255489536.00 +- 82510977335.77 | 202.15 +- 319.21 | 245.15 +- 20.05 |
| XGB | ✗ | ID | 163.83 +- 6.84 | 41.23 +- 1.08 | 448.43 +- 28.97 | 57.45 +- 1.51 | 185.87 +- 2.80 |
| XGB | ✓ | ID | 192.97 +- 9.24 | 40.64 +- 3.07 | 436.29 +- 31.56 | 64.82 +- 1.99 | 189.59 +- 2.78 |
| XGB | ✗ | OD | 95.32 +- 7.37 | 38.19 +- 0.79 | 308.87 +- 21.86 | 56.15 +- 1.34 | 236.05 +- 1.27 |
| XGB | ✓ | OD | 93.56 +- 5.14 | 40.43 +- 3.09 | 304.16 +- 9.07 | 67.22 +- 5.03 | 241.65 +- 3.12 |
| GLU | ✗ | ID | 201.47 +- 1.24 (20.19) | 66.69 +- 1.32 (5.83) | 472.51 +- 43.48 (56.73) | 59.98 +- 0.47 (1.92) | 198.06 +- 4.93 (5.01) |
| GLU | ✗ | OD | 278.74 +- 6.65 (40.23) | 48.10 +- 2.06 (3.13) | 551.14 +- 175.34 (56.50) | 66.74 +- 6.26 (3.37) | 254.05 +- 8.37 (10.83) |
| LAT | ✗ | ID | 206.55 +- 18.61 (75.95) | 39.38 +- 0.55 (1.36) | 405.47 +- 39.58 (68.69) | 50.84 +- 0.59 (1.15) | 183.46 +- 0.70 (4.84) |
| LAT | ✗ | OD | 223.91 +- 21.59 (67.04) | 31.86 +- 0.70 (1.42) | 301.97 +- 104.32 (71.92) | 59.38 +- 1.66 (1.57) | 226.72 +- 14.59 (7.39) |
| NHI | ✗ | ID | 190.20 +- 8.89 (9.07) | 35.20 +- 0.52 (0.26) | 304.60 +- 63.70 (1.16) | 59.02 +- 0.61 (0.27) | 176.60 +- 4.41 (1.95) |
| NHI | ✓ | ID | 262.29 +- 53.94 (26.54) | 82.55 +- 21.48 (6.25) | 926.26 +- 183.16 (31.93) | 66.58 +- 0.73 (1.42) | 179.71 +- 2.55 (0.39) |
| NHI | ✗ | OD | 214.26 +- 6.19 (10.83) | 32.22 +- 0.06 (0.15) | 331.18 +- 128.00 (2.35) | 59.97 +- 3.90 (0.21) | 210.94 +- 9.26 (2.16) |
| NHI | ✓ | OD | 245.25 +- 73.67 (37.93) | 57.18 +- 13.68 (4.05) | 1395.29 +- 755.46 (67.57) | 73.77 +- 2.47 (1.95) | 207.37 +- 13.01 (0.95) |
| TFT | ✗ | ID | 188.64 +- 47.62 (125.97) | 31.58 +- 0.79 (1.50) | 337.31 +- 6.85 (43.82) | 62.66 +- 2.12 (3.63) | 205.19 +- 10.63 (13.71) |
| TFT | ✓ | ID | 215.41 +- 4.50 (32.20) | 42.39 +- 0.36 (0.00) | 339.65 +- 0.71 (35.06) | 70.83 +- 1.80 (3.25) | 224.14 +- 6.01 (5.91) |
| TFT | ✗ | OD | 154.46 +- 32.58 (66.26) | 30.40 +- 1.41 (1.86) | 306.19 +- 58.47 (46.48) | 65.95 +- 6.84 (4.04) | 232.66 +- 26.29 (14.68) |
| TFT | ✓ | OD | 175.62 +- 16.08 (20.39) | 33.53 +- 1.02 (0.00) | 295.58 +- 18.26 (23.85) | 79.73 +- 14.18 (5.57) | 239.46 +- 17.02 (5.88) |
| TRA | ✗ | ID | 228.61 +- 66.99 (0.00) | 41.92 +- 3.73 (0.00) | 276.33 +- 39.70 (0.00) | 62.22 +- 0.99 (0.00) | 174.87 +- 13.02 (0.00) |
| TRA | ✓ | ID | 164.65 +- 12.09 (0.00) | 71.18 +- 0.85 (0.00) | 752.25 +- 154.38 (0.00) | 56.08 +- 1.68 (0.00) | 209.02 +- 9.91 (0.00) |
| TRA | ✗ | OD | 197.03 +- 16.72 (0.00) | 35.65 +- 1.57 (0.00) | 246.66 +- 113.45 (0.00) | 66.89 +- 4.68 (0.00) | 200.23 +- 24.32 (0.00) |
| TRA | ✓ | OD | 189.34 +- 0.39 (0.00) | 52.66 +- 4.41 (0.00) | 1163.43 +- 672.48 (0.00) | 54.77 +- 2.65 (0.00) | 243.19 +- 1.14 (0.00) |

Table 13: Standard error of MAE across data splits and model random initializations.

| | $p(\cdot|x)$ | Data | Broll | Colas | Dubosson | Hall | Weinstock |
|---|---|---|---|---|---|---|---|
| ARI | ✗ | ID | 8.67 ± 0.74 | 4.80 ± 0.04 | 11.06 ± 0.98 | 7.34 ± 0.16 | 11.25 ± 0.10 |
| ARI | ✗ | OD | 9.71 ± 1.93 | 4.87 ± 0.38 | 14.58 ± 4.75 | 6.97 ± 1.19 | 13.34 ± 0.52 |
| LIN | ✗ | ID | 9.71 ± 0.37 | 4.35 ± 0.03 | 9.97 ± 1.00 | 6.33 ± 0.09 | 11.46 ± 0.11 |
| LIN | ✓ | ID | 8.41 ± 0.24 | 4.60 ± 0.04 | 10.03 ± 1.11 | 6.66 ± 0.18 | 11.34 ± 0.10 |
| LIN | ✗ | OD | 9.83 ± 1.58 | 4.41 ± 0.20 | 11.90 ± 3.51 | 6.62 ± 0.91 | 13.09 ± 0.61 |
| LIN | ✓ | OD | 16.80 ± 9.45 | 4.57 ± 0.19 | 67548.59 ± 135072.57 | 10.02 ± 6.68 | 13.16 ± 0.57 |
| XGB | ✗ | ID | 11.50 ± 0.31 | 5.49 ± 0.08 | 19.09 ± 0.32 | 6.55 ± 0.09 | 11.61 ± 0.08 |
| XGB | ✓ | ID | 11.87 ± 0.24 | 5.46 ± 0.21 | 18.55 ± 0.82 | 7.02 ± 0.12 | 11.77 ± 0.16 |
| XGB | ✗ | OD | 8.72 ± 0.45 | 5.32 ± 0.07 | 15.42 ± 0.68 | 6.52 ± 0.11 | 13.04 ± 0.04 |
| XGB | ✓ | OD | 8.56 ± 0.30 | 5.47 ± 0.18 | 15.46 ± 0.35 | 7.11 ± 0.26 | 13.43 ± 0.13 |
| GLU | ✗ | ID | 12.55 ± 0.07 (0.67) | 7.12 ± 0.08 (0.37) | 19.40 ± 1.11 (1.27) | 6.69 ± 0.03 (0.11) | 12.09 ± 0.17 (0.18) |
| GLU | ✗ | OD | 14.82 ± 0.26 (1.09) | 6.03 ± 0.13 (0.21) | 20.70 ± 3.58 (1.05) | 7.04 ± 0.33 (0.18) | 13.65 ± 0.22 (0.32) |
| LAT | ✗ | ID | 12.32 ± 0.60 (2.15) | 5.37 ± 0.04 (0.12) | 17.88 ± 0.85 (1.59) | 6.11 ± 0.04 (0.08) | 11.45 ± 0.01 (0.19) |
| LAT | ✗ | OD | 13.05 ± 0.59 (1.87) | 4.84 ± 0.05 (0.13) | 15.12 ± 2.76 (1.92) | 6.61 ± 0.12 (0.10) | 12.72 ± 0.40 (0.25) |
| NHI | ✗ | ID | 12.07 ± 0.33 (0.31) | 5.04 ± 0.04 (0.02) | 14.79 ± 1.60 (0.05) | 6.57 ± 0.03 (0.02) | 11.21 ± 0.16 (0.07) |
| NHI | ✓ | ID | 14.64 ± 1.57 (0.81) | 8.03 ± 1.40 (0.36) | 27.97 ± 3.06 (0.57) | 7.10 ± 0.03 (0.07) | 11.31 ± 0.09 (0.02) |
| NHI | ✗ | OD | 12.77 ± 0.18 (0.35) | 4.83 ± 0.01 (0.02) | 15.59 ± 3.33 (0.04) | 6.62 ± 0.19 (0.02) | 12.24 ± 0.25 (0.07) |
| NHI | ✓ | OD | 14.01 ± 2.32 (1.03) | 6.65 ± 1.03 (0.27) | 33.52 ± 10.07 (0.83) | 7.53 ± 0.19 (0.11) | 12.12 ± 0.37 (0.03) |
| TFT | ✗ | ID | 11.07 ± 1.17 (2.85) | 4.54 ± 0.05 (0.12) | 15.49 ± 0.05 (1.23) | 6.61 ± 0.12 (0.20) | 11.76 ± 0.35 (0.43) |
| TFT | ✓ | ID | 12.43 ± 0.08 (0.89) | 5.27 ± 0.04 (0.00) | 15.51 ± 0.02 (0.83) | 7.06 ± 0.09 (0.19) | 12.30 ± 0.15 (0.17) |
| TFT | ✗ | OD | 10.23 ± 1.05 (1.91) | 4.47 ± 0.11 (0.15) | 14.53 ± 1.26 (1.11) | 6.76 ± 0.34 (0.21) | 12.50 ± 0.77 (0.45) |
| TFT | ✓ | OD | 11.17 ± 0.55 (0.59) | 4.68 ± 0.12 (0.00) | 14.43 ± 0.36 (0.63) | 7.44 ± 0.65 (0.26) | 12.67 ± 0.43 (0.17) |
| TRA | ✗ | ID | 13.20 ± 2.31 (0.00) | 5.65 ± 0.38 (0.00) | 14.04 ± 1.00 (0.00) | 6.78 ± 0.01 (0.00) | 11.22 ± 0.39 (0.00) |
| TRA | ✓ | ID | 11.27 ± 0.45 (0.00) | 7.77 ± 0.15 (0.00) | 24.40 ± 2.69 (0.00) | 6.42 ± 0.10 (0.00) | 12.61 ± 0.43 (0.00) |
| TRA | ✗ | OD | 12.28 ± 0.83 (0.00) | 5.24 ± 0.28 (0.00) | 12.98 ± 2.91 (0.00) | 7.07 ± 0.14 (0.00) | 11.91 ± 0.72 (0.00) |
| TRA | ✓ | OD | 12.13 ± 0.03 (0.00) | 6.59 ± 0.36 (0.00) | 28.21 ± 8.72 (0.00) | 6.29 ± 0.12 (0.00) | 13.58 ± 0.06 (0.00) |

Table 14: Standard error of likelihood across data splits and model random initializations.

| | $p(\cdot|x)$ | Data | Broll | Colas | Dubosson | Hall | Weinstock |
|---|---|---|---|---|---|---|---|
| ARI | ✗ | ID | -9.93 ± 0.02 | -9.30 ± 0.05 | -10.47 ± 0.35 | -9.81 ± 0.15 | -10.21 ± 0.02 |
| ARI | ✗ | OD | -10.06 ± 0.05 | -9.38 ± 0.04 | -10.44 ± 0.03 | -9.66 ± 0.56 | -10.32 ± 0.05 |
| LIN | ✗ | ID | -9.89 ± 0.01 | -9.19 ± 0.01 | -10.10 ± 0.16 | -9.56 ± 0.03 | -10.14 ± 0.00 |
| LIN | ✓ | ID | -9.87 ± 0.03 | -9.17 ± 0.01 | -10.15 ± 0.17 | -10.30 ± 1.47 | -10.12 ± 0.00 |
| LIN | ✗ | OD | -9.95 ± 0.14 | -9.16 ± 0.06 | -10.11 ± 0.28 | -9.53 ± 0.17 | -10.22 ± 0.03 |
| LIN | ✓ | OD | -10.24 ± 0.30 | -9.16 ± 0.06 | -12.08 ± 3.94 | -10.42 ± 1.49 | -11.13 ± 1.83 |
| XGB | ✗ | ID | -9.94 ± 0.02 | -9.42 ± 0.01 | -10.55 ± 0.02 | -9.68 ± 0.01 | -10.20 ± 0.00 |
| XGB | ✓ | ID | -10.06 ± 0.06 | -9.40 ± 0.02 | -10.54 ± 0.01 | -9.70 ± 0.00 | -10.21 ± 0.00 |
| XGB | ✗ | OD | -10.03 ± 0.01 | -9.36 ± 0.01 | -10.22 ± 0.02 | -9.56 ± 0.01 | -10.28 ± 0.00 |
| XGB | ✓ | OD | -10.03 ± 0.01 | -9.38 ± 0.02 | -10.20 ± 0.01 | -9.53 ± 0.01 | -10.31 ± 0.00 |
| GLU | ✗ | ID | -2.11 ± 0.10 (0.24) | -1.07 ± 0.11 (0.19) | -2.15 ± 0.01 (0.22) | -1.56 ± 0.00 (0.10) | -2.50 ± 0.02 (0.05) |
| GLU | ✗ | OD | -1.96 ± 0.08 (0.27) | -1.61 ± 0.03 (0.12) | -1.17 ± 1.53 (1.69) | -1.44 ± 0.12 (0.11) | -2.41 ± 0.01 (0.05) |
| LAT | ✗ | ID | -25.29 ± 1.96 (5.68) | -10.47 ± 0.01 (0.16) | -52.18 ± 2.17 (9.54) | -20.24 ± 0.39 (0.19) | -26.15 ± 0.03 (0.07) |
| LAT | ✗ | OD | -28.75 ± 2.31 (6.38) | -8.80 ± 0.25 (0.12) | -30.19 ± 3.38 (5.20) | -18.19 ± 0.46 (0.16) | -30.08 ± 1.49 (0.14) |
| NHI | ✗ | ID | -10.01 ± 0.01 (0.01) | -9.32 ± 0.01 (0.00) | -10.37 ± 0.04 (0.00) | -9.62 ± 0.01 (0.00) | -10.13 ± 0.00 (0.00) |
| NHI | ✓ | ID | -10.37 ± 0.07 (0.05) | -9.48 ± 0.10 (0.02) | -10.80 ± 0.01 (0.01) | -9.63 ± 0.01 (0.01) | -10.13 ± 0.00 (0.00) |
| NHI | ✗ | OD | -10.08 ± 0.00 (0.01) | -9.26 ± 0.01 (0.00) | -10.18 ± 0.14 (0.00) | -9.49 ± 0.03 (0.00) | -10.20 ± 0.03 (0.00) |
| NHI | ✓ | OD | -10.21 ± 0.13 (0.04) | -9.36 ± 0.10 (0.01) | -11.10 ± 0.51 (0.04) | -9.58 ± 0.01 (0.01) | -10.19 ± 0.03 (0.00) |
| TRA | ✗ | ID | -9.99 ± 0.09 (0.00) | -9.37 ± 0.04 (0.00) | -10.36 ± 0.04 (0.00) | -9.60 ± 0.03 (0.00) | -10.12 ± 0.00 (0.00) |
| TRA | ✓ | ID | -10.11 ± 0.11 (0.00) | -9.45 ± 0.00 (0.00) | -10.68 ± 0.08 (0.00) | -9.60 ± 0.00 (0.00) | -10.15 ± 0.00 (0.00) |
| TRA | ✗ | OD | -9.98 ± 0.03 (0.00) | -9.30 ± 0.03 (0.00) | -10.09 ± 0.06 (0.00) | -9.47 ± 0.02 (0.00) | -10.17 ± 0.03 (0.00) |
| TRA | ✓ | OD | -10.02 ± 0.01 (0.00) | -9.36 ± 0.01 (0.00) | -10.63 ± 0.27 (0.00) | -9.49 ± 0.02 (0.00) | -10.20 ± 0.03 (0.00) |

Table 15: Standard error of calibration error across data splits and model random initializations.

| | $p(\cdot|x)$ | Data | Broll | Colas | Dubosson | Hall | Weinstock |
|---|---|---|---|---|---|---|---|
| ARI | ✗ | ID | 0.11 ± 0.01 | 0.10 ± 0.01 | 0.10 ± 0.01 | 0.10 ± 0.02 | 0.12 ± 0.01 |
| ARI | ✗ | OD | 0.07 ± 0.02 | 0.08 ± 0.01 | 0.08 ± 0.05 | 0.07 ± 0.01 | 0.12 ± 0.01 |
| LIN | ✗ | ID | 0.12 ± 0.01 | 0.15 ± 0.00 | 0.18 ± 0.02 | 0.10 ± 0.00 | 0.11 ± 0.00 |
| LIN | ✓ | ID | 0.13 ± 0.02 | 0.19 ± 0.01 | 0.21 ± 0.02 | 0.19 ± 0.20 | 0.11 ± 0.00 |
| LIN | ✗ | OD | 0.15 ± 0.04 | 0.15 ± 0.01 | 0.17 ± 0.02 | 0.10 ± 0.02 | 0.11 ± 0.00 |
| LIN | ✓ | OD | 0.55 ± 0.39 | 0.17 ± 0.02 | 0.48 ± 0.58 | 0.23 ± 0.18 | 0.21 ± 0.20 |
| XGB | ✗ | ID | 0.07 ± 0.01 | 0.10 ± 0.00 | 0.07 ± 0.01 | 0.09 ± 0.00 | 0.11 ± 0.00 |
| XGB | ✓ | ID | 0.07 ± 0.01 | 0.09 ± 0.01 | 0.06 ± 0.01 | 0.09 ± 0.00 | 0.10 ± 0.00 |
| XGB | ✗ | OD | 0.11 ± 0.01 | 0.09 ± 0.00 | 0.07 ± 0.01 | 0.08 ± 0.01 | 0.11 ± 0.00 |
| XGB | ✓ | OD | 0.11 ± 0.01 | 0.08 ± 0.01 | 0.07 ± 0.01 | 0.10 ± 0.01 | 0.10 ± 0.00 |
| GLU | ✗ | ID | 0.05 ± 0.01 (0.01) | 0.14 ± 0.01 (0.03) | 0.06 ± 0.00 (0.02) | 0.05 ± 0.00 (0.01) | 0.08 ± 0.00 (0.01) |
| GLU | ✗ | OD | 0.11 ± 0.01 (0.03) | 0.10 ± 0.01 (0.02) | 0.12 ± 0.02 (0.05) | 0.06 ± 0.01 (0.01) | 0.09 ± 0.00 (0.01) |
| LAT | ✗ | ID | 0.36 ± 0.03 (0.05) | 0.25 ± 0.01 (0.03) | 0.42 ± 0.02 (0.03) | 0.30 ± 0.01 (0.02) | 0.33 ± 0.01 (0.02) |
| LAT | ✗ | OD | 0.38 ± 0.01 (0.05) | 0.24 ± 0.02 (0.03) | 0.44 ± 0.11 (0.08) | 0.36 ± 0.04 (0.03) | 0.40 ± 0.01 (0.03) |
| NHI | ✗ | ID | 0.12 ± 0.02 (0.01) | 0.11 ± 0.00 (0.00) | 0.10 ± 0.00 (0.00) | 0.09 ± 0.01 (0.00) | 0.11 ± 0.00 (0.00) |
| NHI | ✓ | ID | 0.07 ± 0.01 (0.01) | 0.21 ± 0.03 (0.04) | 0.08 ± 0.04 (0.02) | 0.07 ± 0.01 (0.00) | 0.11 ± 0.00 (0.00) |
| NHI | ✗ | OD | 0.10 ± 0.00 (0.01) | 0.11 ± 0.00 (0.00) | 0.12 ± 0.01 (0.00) | 0.08 ± 0.01 (0.00) | 0.12 ± 0.00 (0.00) |
| NHI | ✓ | OD | 0.06 ± 0.01 (0.02) | 0.14 ± 0.02 (0.03) | 0.20 ± 0.07 (0.04) | 0.06 ± 0.01 (0.01) | 0.11 ± 0.00 (0.00) |
| TFT | ✗ | ID | 0.16 ± 0.06 (0.08) | 0.07 ± 0.02 (0.03) | 0.23 ± 0.07 (0.10) | 0.07 ± 0.02 (0.02) | 0.07 ± 0.03 (0.03) |
| TFT | ✓ | ID | 0.30 ± 0.08 (0.12) | 0.16 ± 0.08 (0.03) | 0.25 ± 0.03 (0.09) | 0.08 ± 0.01 (0.02) | 0.06 ± 0.03 (0.03) |
| TFT | ✗ | OD | 0.15 ± 0.08 (0.09) | 0.09 ± 0.03 (0.04) | 0.26 ± 0.04 (0.13) | 0.08 ± 0.01 (0.03) | 0.08 ± 0.03 (0.04) |
| TFT | ✓ | OD | 0.23 ± 0.05 (0.10) | 0.09 ± 0.07 (0.02) | 0.35 ± 0.04 (0.10) | 0.08 ± 0.01 (0.02) | 0.05 ± 0.02 (0.02) |
| TRA | ✗ | ID | 0.23 ± 0.07 (0.02) | 0.21 ± 0.09 (0.03) | 0.12 ± 0.01 (0.00) | 0.13 ± 0.01 (0.00) | 0.11 ± 0.01 (0.00) |
| TRA | ✓ | ID | 0.21 ± 0.05 (0.02) | 0.31 ± 0.07 (0.02) | 0.18 ± 0.04 (0.01) | 0.10 ± 0.00 (0.00) | 0.11 ± 0.00 (0.00) |
| TRA | ✗ | OD | 0.19 ± 0.03 (0.01) | 0.22 ± 0.11 (0.04) | 0.14 ± 0.03 (0.01) | 0.15 ± 0.01 (0.00) | 0.12 ± 0.00 (0.00) |
| TRA | ✓ | OD | 0.11 ± 0.03 (0.01) | 0.22 ± 0.06 (0.02) | 0.25 ± 0.09 (0.03) | 0.08 ± 0.01 (0.00) | 0.12 ± 0.00 (0.00) |

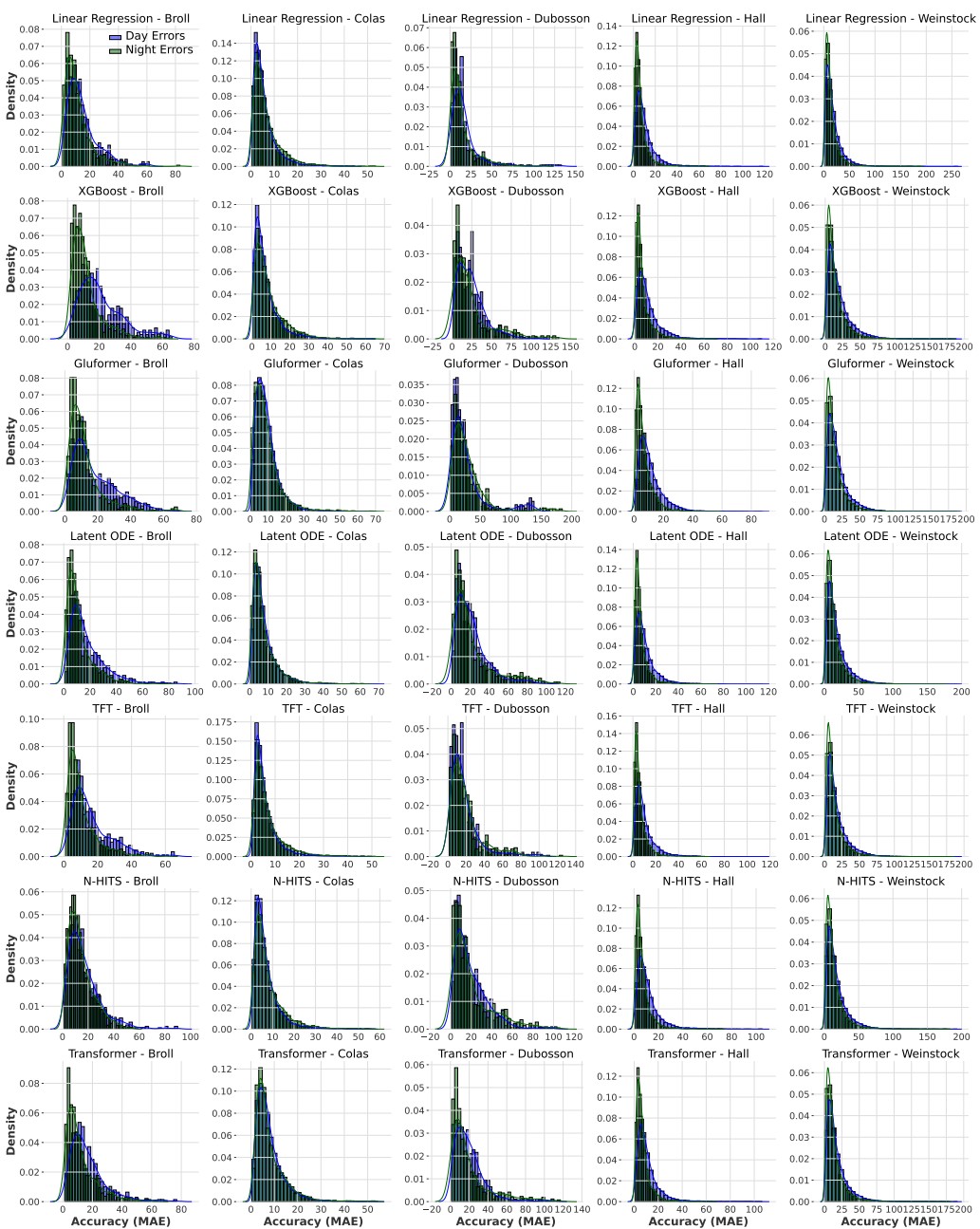

Figure 6: Distribution of daytime (9:00AM to 9:00PM) versus nighttime (9:00PM to 9:00AM) errors (MAE) for models with no covariates on the ID set.

## C    REPRODUCING RESULTS

### C.1    COMPUTE RESOURCES

We conducted all experiments on a single compute node equipped with 4 NVIDIA RTX2080Ti 12GB GPUs. We used Optuna (Akiba et al., 2019) to tune the hyperparameters of all models except ARIMA and saved the best configurations in the `config/` folder of our repository. For ARIMA, we used the native hyperparameter selection algorithm (AutoARIMA) proposed in Hyndman & Khandakar (2008). The search grid for each model is available in the `lib/` folder. The training time varied depending on the model and the dataset. We trained all deep learning models using the Adam optimizer for 100 epochs with early stopping that had a patience of 10 epochs. For AutoARIMA, we used the implementation available in Federico Garza (2022). For the linear regression, XGBoost (Chen & Guestrin, 2016), NHiTS (Challu et al., 2023), TFT (Lim et al., 2021), and Transformer (Vaswani et al., 2017), we used Darts (Herzen et al., 2022) library. For the Gluformer (Sergazinov et al., 2023) and Latent ODE (Rubanova et al., 2019) models, we adapted the original implementation available on GitHub.

The shallow baselines, such as ARIMA, linear regression, and XGBoost, fit within 10 minutes for all datasets. Among the deep learning models, NHiTS was the fastest to fit, taking less than 2 hours on the largest dataset (Weinstock). Gluformer and Transformer required 6 to 8 hours to fit on Weinstock. Latent ODE and TFT were the slowest to fit, taking 10 to 12 hours on Weinstock on average.

### C.2    HYPERPARAMETERS

In this section, we provide an extensive discussion of hyperparameters, exploring their impact on forecasting models' performance across studied datasets. For each model, we have identified the crucial hyperparameters and their ranges based on the paper where they first appeared. We observe that certain models, such as the Latent ODE and TFT, maintain consistent hyperparameters across datasets. In contrast, models like the Transformer and Gluformer exhibit notable variations. We provide a comprehensive list of the best hyperparameters for each datasets in Table 16 and provide intuition below.

**Linear regression and XGBoost (Chen & Guestrin, 2016).** These models are not designed to capture the temporal dependence. Therefore, their hyperparameters change considerably between datasets and do not exhibit any particular patterns. For example, the maximum tree depth of XGBoost varies by 67%, ranging from 6 to 10, while tree regularization remains relatively consistent.

**Transformer (Vaswani et al., 2017), TFT (Lim et al., 2021), Gluformer (Sergazinov et al., 2023).** Both TFT and Gluformer are based on the Transformer architecture and share most of its hyperparameters. For this set of models, we identify the critical parameters to be the number of attention heads, the dimension of the fully-connected layers (absent for TFT), the dimension of the model (hidden_size for TFT), and the number of encoder and decoder layers. Intuitively, each attention head captures a salient pattern, while the fully-connected layers and model dimensions control the complexity of the pattern. The encoder and decoder layers allow models to extract more flexible representations. With respect to these parameters, all models exhibit similar behavior. For larger datasets, e.g. Colas, Hall, and Weinstock, we observe the best performance with larger values of the parameters. On the other hand, for smaller datasets, we can achieve best performance with smaller models.

**Latent ODE (Rubanova et al., 2019).** Latent ODE is based on the RNN (Sutskever et al., 2013) architecture. Across all models, Latent ODE is the only one that consistently shows the best performance with the same set of hyperparameter values, which we attribute to its hybrid nature. In Latent ODE, hyperparameters govern the parametric form of the ODE. Therefore, we believe the observed results indicate that Latent ODE is potentially capturing the glucose ODE.

**NHiTS (Challu et al., 2023).** In the case of NHiTS, its authors identify kernel_sizes as the only critical hyperparameter. This hyperparameter is responsible for the kernel size of the MaxPool operation and essentially controls the sampling rate for the subsequent blocks in the architecture. A larger kernel size leads model to focus more on the low-rate information. Based on our findings, NHiTS selects similar kernel sizes for all datasets, reflecting the fact that all datasets have similar patterns in the frequency domain.

Table 16: Best hyperparameters for each model and dataset selected by Optuna Akiba et al. (2019). For models that support covariates, we indicate best hyperparameters with covariates in parantheses.

| | Hyperparameter | Broll | Colas | Dubosson | Hall | Weinstock |
|---|---|---|---|---|---|---|
| LIN | in_len | 192.00 (12.00) | 12.00 (12.00) | 12.00 (12.00) | 84.00 (60.00) | 84.00 (84.00) |
| XGB | in_len | 84.00 (96.00) | 120.00 (144.00) | 168.00 (36.00) | 60.00 (120.00) | 84.00 (96.00) |
| | lr | 0.51 (0.39) | 0.51 (0.88) | 0.69 (0.65) | 0.52 (0.17) | 0.72 (0.48) |
| | subsample | 0.90 (0.80) | 0.90 (0.90) | 0.80 (0.80) | 0.90 (0.70) | 0.90 (1.00) |
| | min_child_weight | 2.00 (1.00) | 5.00 (3.00) | 5.00 (2.00) | 3.00 (2.00) | 5.00 (2.00) |
| | colsample_bytree | 0.80 (1.00) | 0.90 (0.80) | 0.80 (1.00) | 0.90 (0.90) | 1.00 (0.90) |
| | max_depth | 9.00 (8.00) | 7.00 (5.00) | 10.00 (6.00) | 6.00 (6.00) | 10.00 (6.00) |
| | gamma | 0.50 (1.00) | 0.50 (0.50) | 0.50 (1.50) | 2.00 (1.00) | 0.50 (1.50) |
| | alpha | 0.12 (0.20) | 0.22 (0.06) | 0.20 (0.15) | 0.10 (0.17) | 0.27 (0.16) |
| | lambda_ | 0.09 (0.02) | 0.24 (0.09) | 0.28 (0.09) | 0.13 (0.02) | 0.07 (0.03) |
| | n_estimators | 416.00 (288.00) | 352.00 (416.00) | 416.00 (480.00) | 256.00 (320.00) | 416.00 (320.00) |
| GLU | in_len | 96.00 | 96.00 | 108.00 | 96.00 | 144.00 |
| | max_samples_per_ts | 100.00 | 150.00 | 100.00 | 200.00 | 100.00 |
| | d_model | 512.00 | 384.00 | 384.00 | 384.00 | 512.00 |
| | n_heads | 4.00 | 12.00 | 8.00 | 4.00 | 8.00 |
| | d_fcn | 512.00 | 512.00 | 1024.00 | 1024.00 | 1408.00 |
| | num_enc_layers | 1.00 | 1.00 | 1.00 | 1.00 | 1.00 |
| | num_dec_layers | 4.00 | 1.00 | 3.00 | 1.00 | 4.00 |
| LAT | in_len | 48.00 | 48.00 | 48.00 | 48.00 | 48.00 |
| | max_samples_per_ts | 100.00 | 100.00 | 100.00 | 100.00 | 100.00 |
| | latents | 20.00 | 20.00 | 20.00 | 20.00 | 20.00 |
| | rec_dims | 40.00 | 40.00 | 40.00 | 40.00 | 40.00 |
| | rec_layers | 3.00 | 3.00 | 3.00 | 3.00 | 3.00 |
| | gen_layers | 3.00 | 3.00 | 3.00 | 3.00 | 3.00 |
| | units | 100.00 | 100.00 | 100.00 | 100.00 | 100.00 |
| | gru_units | 100.00 | 100.00 | 100.00 | 100.00 | 100.00 |
| NHI | in_len | 96.00 (144.00) | 132.00 (96.00) | 108.00 (120.00) | 144.00 (120.00) | 96.00 (96.00) |
| | max_samples_per_ts | 50.00 (50.00) | 100.00 (50.00) | 50.00 (50.00) | 100.00 (50.00) | 200.00 (50.00) |
| | kernel_sizes | 5.00 (3.00) | 3.00 (3.00) | 3.00 (2.00) | 4.00 (5.00) | 4.00 (3.00) |
| | dropout | 0.13 (0.09) | 0.18 (0.13) | 0.06 (0.16) | 0.05 (0.19) | 0.13 (0.10) |
| | lr | 0.00 (0.00) | 0.00 (0.00) | 0.00 (0.00) | 0.00 (0.00) | 0.00 (0.00) |
| | batch_size | 64.00 (32.00) | 32.00 (48.00) | 32.00 (48.00) | 48.00 (48.00) | 64.00 (32.00) |
| | lr_epochs | 16.00 (10.00) | 2.00 (16.00) | 2.00 (12.00) | 2.00 (4.00) | 16.00 (2.00) |
| TFT | in_len | 168.00 (96.00) | 132.00 (120.00) | 168.00 (120.00) | 96.00 (132.00) | 132.00 (108.00) |
| | max_samples_per_ts | 50.00 (50.00) | 200.00 (100.00) | 50.00 (50.00) | 50.00 (50.00) | 200.00 (50.00) |
| | hidden_size | 80.00 (80.00) | 256.00 (32.00) | 240.00 (240.00) | 160.00 (64.00) | 96.00 (112.00) |
| | num_attention_heads | 4.00 (3.00) | 3.00 (3.00) | 2.00 (1.00) | 2.00 (3.00) | 3.00 (2.00) |
| | dropout | 0.13 (0.23) | 0.23 (0.11) | 0.25 (0.24) | 0.13 (0.15) | 0.14 (0.15) |
| | lr | 0.00 (0.01) | 0.00 (0.00) | 0.00 (0.00) | 0.00 (0.00) | 0.00 (0.00) |
| | batch_size | 32.00 (32.00) | 32.00 (32.00) | 64.00 (32.00) | 48.00 (32.00) | 48.00 (48.00) |
| | max_grad_norm | 0.53 (0.03) | 0.98 (0.80) | 1.00 (0.09) | 0.43 (0.66) | 1.00 (0.95) |
| TRA | in_len | 96.00 (108.00) | 108.00 (120.00) | 108.00 (156.00) | 144.00 (132.00) | 96.00 (96.00) |
| | max_samples_per_ts | 50.00 (50.00) | 200.00 (200.00) | 50.00 (50.00) | 200.00 (150.00) | 50.00 (50.00) |
| | d_model | 96.00 (128.00) | 64.00 (128.00) | 32.00 (64.00) | 64.00 (64.00) | 128.00 (128.00) |
| | n_heads | 4.00 (2.00) | 2.00 (4.00) | 2.00 (2.00) | 4.00 (4.00) | 2.00 (4.00) |
| | num_encoder_layers | 4.00 (2.00) | 3.00 (4.00) | 1.00 (2.00) | 1.00 (1.00) | 2.00 (1.00) |
| | num_decoder_layers | 1.00 (2.00) | 3.00 (1.00) | 1.00 (1.00) | 1.00 (3.00) | 4.00 (4.00) |
| | dim_feedforward | 448.00 (160.00) | 480.00 (128.00) | 384.00 (384.00) | 96.00 (192.00) | 64.00 (448.00) |
| | dropout | 0.10 (0.04) | 0.12 (0.20) | 0.04 (0.00) | 0.01 (0.13) | 0.00 (0.19) |
| | lr | 0.00 (0.00) | 0.00 (0.00) | 0.00 (0.00) | 0.00 (0.00) | 0.00 (0.00) |
| | batch_size | 32.00 (32.00) | 32.00 (32.00) | 32.00 (48.00) | 48.00 (48.00) | 32.00 (48.00) |
| | lr_epochs | 16.00 (20.00) | 8.00 (18.00) | 6.00 (20.00) | 14.00 (4.00) | 4.00 (4.00) |
| | max_grad_norm | 0.67 (0.89) | 0.83 (0.82) | 0.21 (0.10) | 0.43 (0.23) | 0.42 (0.19) |

## D  CHALLENGES

In addressing the challenges associated with the implementation of our predictive models in clinical settings, we recognize three pivotal obstacles. Firstly, the challenge posed by computing power necessitates a strategic refinement of our models to guarantee their effectiveness on devices grappling with resource limitations and potential disruptions in internet connectivity. The delicate balance between the complexity of the model and its real-time relevance emerges as a critical factor, especially within the dynamic contexts of diverse healthcare settings.

Secondly, the challenge of cold starts for new enrolling patients presents a significant hurdle. We acknowledge the importance of devising strategies to initialize and tailor the predictive models for individuals who are newly enrolled in the system. This consideration underscores the need for a dynamic and adaptable framework that ensures the seamless integration of our models into the continuum of patient care.

The third challenge pertains to data privacy and transmission. To address this, our models must either possess on-device training capabilities or facilitate the secure and anonymized transmission of data to external servers. This emphasis on safeguarding patient information aligns with contemporary standards of privacy and ethical considerations, reinforcing the responsible deployment of our models in clinical practice.

