# OpenReview forum: "GlucoBench: Curated List of Continuous Glucose Monitoring Datasets with Prediction Benchmarks"
_ICLR.cc/2024/Conference — ICLR 2024 poster_

### Official Review · Reviewer_XK2E · 2023-10-30

**Soundness:** 2 fair
**Presentation:** 2 fair
**Contribution:** 1 poor
**Rating:** 5
**Confidence:** 4

**Summary:**

This paper presents a curated list of continuous glucose monitoring (CGM) datasets with a list of standard tasks for the problem of CGM both in terms of predictive accuracy and uncertainty quantification. It also provides benchmarking results using classic to modern methods on the datasets. Detailed analysis and discussion of the impacts of different factors on performance are provided.

**Strengths:**

1. The authors provide systematic benchmark results of different CGM models on multiple CGM datasets. Their evaluation methods seem to be reproducible and fair.
2. They provide diverse analyses of the benchmarking results, including the impact of dataset size, generalization of the models, and impact of time of day.

**Weaknesses:**

1. The writing of some parts is not clear and easy to understand. For example, when describing Task 2, it is not clear what “uncertainty” actually means here.; the caption of Figure 3 is also not clear enough about what the “green block” refers to with more than one green element in the figure; on Page 8, the authors did not provide an exact definition of in-distribution and out-of-distribution test, making the readers have to guess what it means by the authors.
2. I am concerned about the level of contribution of this work. Many of the conclusions of the paper are not novel. For example, people should already know “Healthy subjects demonstrate markedly smaller errors” and “using simpler shallow models when data is limited”.

**Questions:**

1. What is the motivation for some analysis you did in the paper? For example, what is the motivation for exploring the impact of time of day on accuracy? Why should researchers care about the analysis you did in Section 5, especially under the context of CGM?
2. How is this work different from other review papers mentioned in related works?

---

> ### Author Response · Authors · 2023-11-16
> **Our thanks and reply to reviewer XK2E**
>
> > The writing of some parts is not clear and easy to understand. For example, when describing Task 2, it is not clear what “uncertainty” actually means here.; the caption of Figure 3 is also not clear enough about what the “green block” refers to with more than one green element in the figure; on Page 8, the authors did not provide an exact definition of in-distribution and out-of-distribution test, making the readers have to guess what it means by the authors.
>
> Thank you for bringing our attention to this! We certainly want to make the manuscript accessible.
> - By uncertainty quantification, we mean measure of the distribution fit of the predictive models to the data. For this purposes we use log-likelihood and regression calibration [1] for distribution-free models (e.g. methods bases on quantile regression). We add this description to p.2, where we introduce the task.
> - By "green blocks" in Figure 3, we mean the small green rectangles, which point out to the locations in the sequence where the values have been interpolated. We have modified the caption of Figure 3 to avoid confusion.
> - If any part of the data for a patient was used for training, we call the corresponding test set "in-distribution." If the data for a patient was completely withheld from the model at all stages (training / validation), then we call such test set "out-of-distribution." We introduce this notation in Section 4.3 on p.6. For readers' convenience, we also add clarifications on p.8.
>
> We highlight all our edits in the revised manuscript in red.
>
> > I am concerned about the level of contribution of this work. Many of the conclusions of the paper are not novel. For example, people should already know “Healthy subjects demonstrate markedly smaller errors” and “using simpler shallow models when data is limited”.
>
> Medical facts may not be immediate for ML practitioners and vice versa for medical researchers. Therefore, part of the novelty of our work is in empirically establishing intuitive conclusions (e.g. errors are smaller for healthy subjects and shallow models are better for small data) and debunking popular misconceptions in the specific context of CGM. For example, it is common to assume that for large datasets deep learning must be a default choice. However, we show in Section 5.1 on the example of Colas (2nd largest dataset) that when there is less 7 days of data per patient, the deep learning models are unable to close the performance gap with the shallow -- the fact that has never been established in the literature before.
>
> > What is the motivation for some analysis you did in the paper? For example, what is the motivation for exploring the impact of time of day on accuracy? Why should researchers care about the analysis you did in Section 5, especially under the context of CGM?
>
> Our motivation for writing Section 5 is to help guide intuition for future model development and benchmarking based on the empirical facts. In each subsection of the analysis, our goal was to highlight a major shortcoming of the current models and their evaluation methods and contribute a discussion on the future steps. For example, in Section 5.1, we highlight that a patient's stochastic behavior poses a major challenge to all forecasting models. At the same time, we observe that each patient tends to adhere to a schedule with repetitive pronounced states, such as active / rest cycles, which affect the models' accuracy. Therefore, when designing new experiments, we need to control data splitting to account for different times of the day.
>
> > How is this work different from other review papers mentioned in related works?
>
> Previous works can be divided into review [2, 3, 4, 5] and benchmarking papers [6, 7]. Review papers do not explicitly compare the models and do not have an associated code base, but rather offer a survey of the literature with a summary of existing methods. Previous benchmarking papers [6, 7] have focused on a single dataset, excluded hybrid models from the analysis, and compared the model solely based on their accuracy.
>
> [1] V. Kuleshov, et al. Accurate uncertainties for deep learning using calibrated regression. ICML, 2018.
>
> [2] S. Oviedo et al. A review of personalized blood glucose prediction strategies for T1DM patients. IJNMBE, 2017.
>
> [3] I. Contreras et al. Artificial Intelligence for Diabetes Management and Decision Support: Literature Review. JMIR, 2018.
>
> [4] H. Kim et al. Lessons from Use of Continuous Glucose Monitoring Systems in Digital Healthcare. JEM, 2020.
>
> [5] I. Kavakiotis, et al. Machine Learning and Data Mining Methods in Diabetes Research. CSBJ, 2017.
>
> [6] S. Mirshekarian, et al. LSTMs and Neural Attention Models for Blood Glucose Prediction: Comparative Experiments on Real and Synthetic Data. IEEE EMBS, 2019.
>
> [7] J. Xie et al. Benchmarking Machine Learning Algorithms on Blood Glucose Prediction for Type I Diabetes in Comparison With Classical Time-Series Models. IEEE TBE, 2020.

---

> > ### Comment · Reviewer_XK2E · 2023-11-23
> > **Thanks for the response.**
> >
> > I appreciate the clarification. I will keep my score.

---

### Official Review · Reviewer_HdpX · 2023-10-30

**Soundness:** 2 fair
**Presentation:** 2 fair
**Contribution:** 2 fair
**Rating:** 6
**Confidence:** 3

**Summary:**

The manuscript analyses continuous glucose monitoring (CGM) data and glucose prediction models for diabetes management. It addresses the challenges associated with accurate glucose prediction, evaluates various models, and offers a curated collection of public CGM datasets. The paper focuses on two main tasks: (1) enhancing predictive accuracy and (2) improving uncertainty quantification in glucose prediction. It discusses the impact of dataset size, patient composition, and covariates on model performance and generalization. The study emphasizes the need for cautious use of glucose predictions in diabetes management and proposes future research directions.

**Strengths:**

1) The paper addresses the important issue of model generalization to new patient populations, providing evidence of the challenges associated with individual-level variation.
2) The key strength of the manuscript is it presents a valuable resource for the diabetes research community by offering public CGM datasets, standardized tasks, benchmark models, and detailed performance analysis. It also provides valuable insights into the factors affecting glucose prediction, including dataset size, patient populations, and time of day. It highlights the varying impact of covariates on model performance across different datasets.
3) I believe the study follows rigorous principles of reproducibility, ensuring that results are consistent across different data splits and model re-runs. It also focuses on fair comparisons by considering out-of-the-box model performance.

**Weaknesses:**

Some terms and concepts in the manuscript, such as ARIMA, Latent ODE, and Bayesian optimization, may be challenging for readers without a deep background in machine learning and statistics. Also, the choice of benchmark models might not be exhaustive, and the paper could benefit from a more extensive discussion of the models' suitability for different scenarios.

**Questions:**

I recommend authors consider below suggestions/limitations:-
1) Are there any potential biases in the selection of the public datasets, and how might they impact the generalizability of the results? Authors should discuss about the quality of the selected CGM datasets assessed, and what criteria were used to determine data quality.
2) The paper discusses the impact of covariates on model performance. Can you provide insights into which specific covariates had the most influence on the predictions and whether their quality was uniform across all datasets? And, how does the decision to omit certain models due to their lack of support for covariates affect the overall model comparison?
3) Perhaps authors should consider including, at least, in the supplemental material about any potential risks or challenges in directly applying the findings from the study to clinical settings especially when translated into practical clinical applications for healthcare providers and individuals with diabetes.
4) The paper mentions not considering additional model-specific tuning for benchmark models. Could such optimizations significantly enhance the models' performance, and why were they excluded from the evaluation?
5)  How diverse were the patient populations in the selected datasets in terms of demographics, disease severity, or other relevant factors, and how might this diversity affect the results and generalization?

---

> ### Author Response · Authors · 2023-11-16
> **Our thanks and reply to reviewer HdpX**
>
> Thank you for all of the constructive feedback! We believe that your feedback enabled us to significantly improve the paper. Please find our response to your questions below.
>
> > Are there any potential biases in the selection of the public datasets, and how might they impact the generalizability of the results?
>
> We recognize that using publicly available datasets may introduce biases into the analysis arising from (1) selection bias, e.g., vulnerable sub-populations incapable of accepting public information sharing agreement are excluded (2) missing information due to privacy concerns, e.g., demographic information or medical history may be partially concealed (3) conflict of interest, e.g., researchers may be funded by external sources to collect and publish data that supports certain claims etc. Previous works such as [1, 2] that have focused on developing benchmarks have only included analyzed a single dataset. Therefore, to help offset the possible biases, we have included 5 datasets collected by medical professionals from different countries in our benchmark: Broll (USA), Colas (Spain), Dubosson (France), Hall (USA), and Weinstock (USA). At the same time, we note that our efforts in developing a more representative benchmark are ongoing. We welcome contributions and will make our code available online at github.com/XXX.
>
> > Authors should discuss about the quality of the selected CGM datasets assessed, and what criteria were used to determine data quality.
>
> We thank the reviewer from bringing this point up! Data quality is crucial in devising benchmarks, and we have paid meticulous attention to ensuring it. On CGM curve level, we ensure that the measurements remain in range (between 20 mg/dL and 400 mg/dL), do not exhibit drastic fluctuations > 40 mg/dL in 5 minutes), and are not constant. On the patient level, we check that each patient has enough data (> 16 hours of continuous recording). Finally, on the dataset level, we ensure that there are enough patients to allow for in- / out-of-distribution splits (> 5 patients). We emphasize these details in Section 3.1 and highlight the changes in red.
>
> > The paper discusses the impact of covariates on model performance. Can you provide insights into which specific covariates had the most influence on the predictions and whether their quality was uniform across all datasets?
>
> We provide summary of the covariates for each dataset in Table 8 in the Supplement and include it here for reference. Besides demographic information, each dataset is unique in the set of provided covariates. We ensure the quality by checking each covariate distribution, where we check for range and any possible data corruption.
>
> Out of 5 models that support covariates, only XGBoost consistently (on 3 out of 5 datasets) benefits from their inclusion. For XGBoost, we report the covariate importance in Table 11 in the Supplement. In general, we observe that:
>
> - Among features available for all datasets, time features such as hour and day of the week, consistently appear in the top 3 important features across all datasets, indicating that patients possibly tend to adhere to daily routines that affect their glucose patterns.
> - Dynamic physical activity features such as heart rate and blood pressure are only available for Dubosson and appear to be all selected as highly important.
> - Demographic and medical record information is not available for Broll and Dubosson. For the rest of the datatsets, we observe medication (e.g. Vitamin D, Lisinopril for Weinstock), disease indicators (e.g. Diabetes T2 for Colas, Osteoporosis for Weinstock), health summary metrics (Body Mass Index for Colas), as well as indices derived from CGM measurements (e.g. J-index) being selected as highly important.
>
> > How does the decision to omit certain models due to their lack of support for covariates affect the overall model comparison?
>
> We compare models with and without covariates separately and do not exclude the latter from the overall analysis.
>
> [1] S. Mirshekarian, H. Shen, R. Bunescu, and C. Marling. LSTMs and Neural Attention Models for Blood Glucose Prediction: Comparative Experiments on Real and Synthetic Data. Annual International Conference of the IEEE Engineering in Medicine and Biology Society. IEEE Engineering in Medicine and Biology Society., 2019:706–712, 2019.
>
> [2] J. Xie and Q. Wang. Benchmarking Machine Learning Algorithms on Blood Glucose Prediction for Type I Diabetes in Comparison With Classical Time-Series Models. IEEE Transactions on Biomedical Engineering, 67(11):3101–3124, 2020.

---

> > ### Comment · Reviewer_HdpX · 2023-12-03
> >
> > Thank you for providing more clarification! I changed my rating, good luck!

---

> ### Author Response · Authors · 2023-11-16
> **Reply to reviewer HdpX (continued)**
>
> > Perhaps authors should consider including, at least, in the supplemental material about any potential risks or challenges in directly applying the findings from the study to clinical settings especially when translated into practical clinical applications for healthcare providers and individuals with diabetes.
>
> We include a discussion of the potential negative impact in Section 6. Implementing any of the predictive models in automatic or recommender systems without checks or proper education on its shortcomings may be lethally dangerous. In an automatic system (such as Aritifical Pancreas), if a model makes an error and the system overcompensates with insulin, this could lead to serious health consequences. Similarly, without proper education on the model shortcomings, a person may believe mistaken model predictions and also overcompensate with insulin. Apart from these more obvious risks from direct model or system malfunction, there are potential risks from device hacking or intentional harm. Therefore, implementing the models in clinical practice requires careful consideration and multiple defense levels.
>
> In terms of the challenges, there is a computing power challenge since any of the models must be able to run on the device with possibly no internet connection. Second, there is a challenge with cold starts for new enrolling patients. Third, there is data privacy and transmission challenge, as the model must be either able to train on the device itself or the data must be transmitted to the server in anonymized way. We add this discussion to Section 6 and the Supplement (Appendix D) and highlight our changes in red.
>
> > The paper mentions not considering additional model-specific tuning for benchmark models. Could such optimizations significantly enhance the models' performance, and why were they excluded from the evaluation?
>
> In tuning the models, we have followed the original authors' suggestions on the best set of tuning parameters. We summarize the tuning parameters for each model in Table 16 in the Supplement. However, we do not go further than the author suggested best practices on tuning the models. As an example, for the Transformer, we use the vanilla implementation with encoder-decoder structure as proposed in [3], and do not investigate further possible developments that enhance input length capabilities or expressiveness of the model as proposed in [4, 5, 6]. Such developments indeed may boost the base model performance but should be included as a separate entry. Indeed, we are actively adding new base models and hope that publication will add more light to our effort and bring attention to this crucial task.
>
> > How diverse were the patient populations in the selected datasets in terms of demographics, disease severity, or other relevant factors, and how might this diversity affect the results and generalization?
>
> One of the main motivations of our paper was to establish benchmarks on a diverse set of datasets covering multiple subpopulations. We summarize each dataset demographic composition in Table 2 in the main paper and provide it here for reference. With an exception of Hall, all patients were recruited during routine clinic visits: diabetes clinics for Dubosson (France), Weinstock (USA), and Broll (USA), and hypertension and vascular risk clinic for Colas (Spain). For Hall (USA), human subjects were recruited via newspapers and community outreach. Based on the table and recruiting patterns, we observe good coverage across various age, gender, and disease status groups. Therefore, we believe our study is reasonably randomized and representative. However, we note that our efforts are ongoing and we encourage and solicit the addition of new datasets.
>
> [3] A. Vaswani, N. Shazeer, N. Parmar, J. Uszkoreit, L. Jones, A. N. Gomez, Ł . Kaiser, and I. Polosukhin. Attention is all you need. Advances in Neural Information Processing Systems (NeurIPS), 30, 2017.
>
> [4] S. Liu, H. Yu, C. Liao, J. Li, W. Lin, A. X. Liu, and S. Dustdar. Pyraformer: Low-complexity pyramidal attention for long-range time series modeling and forecasting. In International conference on learning representations, 2021.
>
> [5] H. Wu, J. Xu, J. Wang, and M. Long. Autoformer: Decomposition transformers with auto-correlation for long-term series forecasting. Advances in Neural Information Processing Systems, 34:22419–22430, 2021
>
> [6] H. Zhou, S. Zhang, J. Peng, S. Zhang, J. Li, H. Xiong, and W. Zhang. Informer: Beyond efficient transformer for long sequence time-series forecasting. In Proceedings of the AAAI conference on artificial intelligence, volume 35, pages 11106–11115, 2021.

---

> ### Author Response · Authors · 2023-11-16
> **Reply to reviewer HdpX (continued 2)**
>
> **Table 2 in the main paper**: Demographic information (average) for each dataset before (Raw) and after pre-processing
> (Processed). CGM indicates the device type; all devices have 5 minute measurment frequency.
>
> |           | Diabetes | Device        | # of subjects | # of subjects | Age | Age       | Sex       | Sex       |
> |-----------|----------|---------------|---------------|---------------|-----|-----------|-----------|-----------|
> | Dataset   | Overall  | Overall       | Raw           | Processed     | Raw | Processed | Raw       | Processed |
> | Broll     | Type 2   | Dexcom G4     | 5             | 5             | NA  | NA        | NA        | NA        |
> | Colas     | Mixed    | MiniMed iPro  | 208           | 201           | 59  | 59        | 103 / 104 | 100 / 100 |
> | Dubosson  | Type 1   | MiniMed iPro2 | 9             | 7             | NA  | NA        | 6 / 3     | NA        |
> | Hall      | Mixed    | Dexcom G4     | 57            | 56            | 48  | 48        | 25 / 32   | NA        |
> | Weinstock | Type 1   | Dexcom G4     | 200           | 192           | 68  | NA        | 106 / 94  | 101 / 91  |
>
> **Table 8 in the supplement**: Covariate information for each dataset.
>
> |                 | Covariate  | Broll | Colas | Dubosson | Hall | Weinstock |
> |------------------|------------|-------|-------|----------|------|-----------|
> | Static          | Age        |       | x     |          | x    |           |
> |                 | Height     |       |       |          | x    | x         |
> | Total #          |            | 0     | 7     | 0        | 48   | 38        |
> | Dynamic-known   | Year       | x     | x     | x        | x    | x         |
> |                 | Month      | x     | x     | x        | x    | x         |
> | Total #          |            | 6     | 5     | 5        | 5    | 5         |
> | Dynamic-unknown | Insulin    |       |       | x        |      |           |
> |                 | Heart Rate |       |       | x        |      |           |
> | Total #           |            | 0     | 0     | 11       | 0    | 0         |
>
> **Table 11 in the supplement**: Top-10 features with importance weights selected by XGBoost for each dataset.
> | Broll       | Broll      | Colas                 | Colas      | Dubosson                     | Dubosson   | Hall                         | Hall       | Weinstock                  | Weinstock  |
> |-------------|------------|-----------------------|------------|------------------------------|------------|------------------------------|------------|----------------------------|------------|
> | Covariate   | Importance | Covariate             | Importance | Covariate                    | Importance | Covariate                    | Importance | Covariate                  | Importance |
> | Month       | 0.001428   | Hour                  | 0.000634   | Slow Insulin Intake          | 0.000625   | Day of week                  | 0.002322   | Minute                     | 0.000444   |
> | Day of week | 0.001144   | Day of week           | 0.000202   | Hour                         | 0.000390   | Median CGM                   | 0.002044   | Day of week                | 0.000365   |
> | Second      | 0.000886   | Glycemia              | 0.000133   | heart Rate Variability Index | 0.000359   | J Index of CGM               | 0.001808   | Hour                       | 0.000291   |
> | Hour        | 0.000768   | Minute                | 0.000126   | Body Temperature             | 0.000323   | Hour                         | 0.001786   | Vitamin D                  | 0.000197   |
> | Minute      | 0.000410   | Diabetes T2           | 0.000121   | Posture                      | 0.000296   | Freq. High CGM               | 0.001576   | Year                       | 0.000194   |
> | Patient ID  | 0.000072   | Patient ID            | 0.000104   | Activity                     | 0.000292   | Freq. Low CGM                | 0.001153   | Erectile dysfunction       | 0.000154   |
> | Year        | 0.000000   | \# Follow Up Visits   | 0.000093   | Calories                     | 0.000291   | Minute                       | 0.001136   | Osteoporosis               | 0.000140   |
> |             |            | Body Mass Index (BMI) | 0.000090   | Heart Rate                   | 0.000197   | \% Pre-Diabetec CGM          | 0.001054   | Chronic kidney disease     | 0.000140   |
> |             |            | Age                   | 0.000085   | Blood Pressure               | 0.000190   | Coefficient of CGM Variation | 0.000779   | \# of Meter Checks per Day | 0.000137   |
> |             |            | Gender                | 0.000070   | Fast Insulin Intake          | 0.000140   | Variance of CGM              | 0.000706   | Lisinopril                 | 0.000128   |

---

### Official Review · Reviewer_5DZS · 2023-11-01

**Soundness:** 3 good
**Presentation:** 4 excellent
**Contribution:** 4 excellent
**Rating:** 8
**Confidence:** 4

**Summary:**

The paper demonstrates the initiative and efforts in creating a public benchmark data repository for research in continuous glucose monitoring (CGM), with a curated collection of diverse CGM datasets, popular data tasks, bench-marking protocol and baseline models performance comparisons from existing literature. Code repository is available as supplementary material.

**Strengths:**

- Originality: the paper demonstrates an initiative in public data repository of ML application in CMG research
- Significance: The initiative of creating a public data repository of CGM research documented in the paper is of great value to the both clinical and ML research communities for experiment reproduction, bench-marking new methods and potential application adoption.
- Quality: The paper is well-written with inspiring research questions. Model comparisons are performed on multiple datasets with inspiring research questions and discussion on results in part 5.
- Clarity: The paper is well-organized with problem formulation, related work, dataset and data tasks description, benchmarking protocols and detailed discussion.
- Code repository is provided with the paper submission

**Weaknesses:**

- Cross validation results will be preferred than simple train/test/validation splitting in benchmark model performance comparison.
- Automated machine learning (AutoML) could also help in benchmarking performance across all datasets with more ML models and pre-processing pipelines.

**Questions:**

1. How will new datasets be added to the repository?
2. How could other researchers contribute to the repository?

---

> ### Author Response · Authors · 2023-11-16
> **Our thanks and reply to reviewer 5DZS**
>
> Thank you for all of the constructive comments! We sincerely appreciate the thoroughness and detail that went into your review. Please, find our reply to your questions below.
>
> > Cross-validation results will be preferred than simple train/test/validation splitting in benchmark model performance comparison.
>
> We agree that a simple train/validation/test split is not sufficient for accurately evaluating the model. In fact, we note in Section 5.1, how different splits of the data based on the time of the day may drastically affect the model performance. However, we found that full cross-validation, e.g. 10-fold cross-validation, is too computationally demanding for some methods. Thus, as a compromise, we use 2 random splits (rather than just one split) to evaluate the performance, which we describe in Section 4.3. We report the standard deviation of the errors for both tasks across random data splits in the Tables 12-15 in the Supplement.
>
> > Automated machine learning (AutoML) could also help in benchmarking performance across all datasets with more ML models and pre-processing pipelines.
>
> Thank you for the suggestion! Currently, we are using Optuna [1] to automate model hyper-parameter optimization. As a future step, we recognize that incorporating a fully automated pre-processing, training, and evaluation pipeline would significantly simplify the use of our benchmark and increase its adoption.
>
> > How will new datasets be added to the repository?
>
> By design, we have made adding new datasets convenient. To add a new dataset, there are 2 steps: (1) uploading the data, (2) specifying dataset information. Our code expects the dataset to be specified as a table in .csv format with columns for date, unique patient identifier, and any additional covariates. All the information about the dataset, e.g. sampling rate, covariate names, types, and data types, must be specified in the separate .yaml file. We provide detailed examples with instructions on how to add a new dataset in the exploratory analysis folder of the code, which we will make available at github.com/XXX.
>
> > How could other researchers contribute to the repository?
>
> This is an excellent question! There are 3 ways to contribute: (1) adding a new dataset, (2) adding a new model, (3) helping build out the code base.  The code will be available online upon the publication of the paper at github.com/XXX. We welcome any contribution and are soliciting help from researchers working on CGM. In the revised paper, we add a link to the online repository to the Abstract and Section 6, where we discuss future directions.
>
> [1] T. Akiba, S. Sano, T. Yanase, T. Ohta, and M. Koyama. Optuna: A next-generation hyper-parameter optimization framework. In Proceedings of the 25th ACM SIGKDD international conference on knowledge discovery & data mining, pages 2623–2631, 2019.

---

### Author Response · Authors · 2023-11-16
**Revised the paper according to the suggestions from the reviewers**

We are grateful for the insightful feedback provided by the reviewers! Incorporating their suggestions has strengthened the manuscript considerably. We have uploaded the revised main paper and supplement, with all edits highlighted in red to indicate the revisions made. We provide a summary of revisions for convenience:
1. (Answering to reviewer 5DZS) Added placeholder links to our online code base in the Abstract on p.1 and Section 6 on p.9.
2. (Answering to reviewer HdpX) Added a discussion on the quality of the data in Section 3.1 on p.3.
3. (Answering to reviewer HdpX) Expanded the discussion on the potential risks in Section 6 on p. 9 and the Supplement (Appendix D).
4. (Answering to reviewer HdpX) Clarified Figure 3 caption on p. 4.
5. (Answering to reviewer XK2E) Clarified the uncertainty quantification task description on p.2 in Section 1.
6. (Answering to reviewer XK2E) Provided more details on in- / out-of-distribution data split on p.8 in Section 5.2.

We believe these changes address the reviewers' valuable recommendations and hope you find the paper improved. Please let us know if any additional modifications would further enhance the work. We appreciate you taking the time to consider our revisions and look forward to hearing your thoughts.

---

### Meta-Review · Area_Chair_gCDf · 2023-12-10

**Metareview:**

This submission presents "GlucoBench," a comprehensive benchmark data repository for continuous glucose monitoring (CGM) research. This work includes a curated collection of CGM datasets, popular data tasks, benchmarking protocols, and baseline model performance comparisons. The authors aim to facilitate research in CGM by providing resources for experiment reproduction, benchmarking new methods, and potential application adoption. The submission also includes a code repository.

Reviewer 5DZS appreciates the originality and significance of the paper, highlighting its value to both clinical and ML research communities. The paper is praised for its clarity, organization, and the inclusion of a code repository. However, the reviewer suggests improvements in benchmark model performance comparison, such as using cross-validation and incorporating Automated Machine Learning (AutoML). Questions about the addition of new datasets and community contributions to the repository were raised.  Reviewer HdpX acknowledges the paper's contribution to understanding model generalization in CGM and the provision of valuable resources for diabetes research. The paper is noted for its focus on reproducibility and fair model comparisons. However, concerns are raised about the accessibility of some technical content and the choice of benchmark models. The reviewer suggests discussing potential biases in dataset selection, the impact of covariates, and the implications of applying findings to clinical settings. Reviewer XK2E points out issues with clarity in the writing and questions the novelty of the work's conclusions. Concerns are raised about the motivation behind certain analyses and how this work differs from other review papers. The authors provided clarifications, which were acknowledged by Reviewer XK2E, but the score remained unchanged.

The paper seems a valuable contribution to the field of CGM research, providing a significant resource for the community. The strengths of the paper lie in its original clinical applications, the comprehensive nature of the dataset collection, and the benchmarking efforts. The inclusion of a code repository enhances the paper's utility and reproducibility. However, there are areas for improvement. The paper could benefit from a more detailed discussion on the methodology, particularly in terms of model selection and the impact of covariates. Addressing the clarity issues raised by Reviewer XK2E would also strengthen the paper. The concerns about the novelty of the conclusions and the motivation behind certain analyses need to be addressed to establish the paper's contribution more firmly.

Overall, considering the mixed reviews and the scores provided, the paper is on the borderline of acceptance. The decision should weigh the paper's significant contributions and resources against the concerns raised about its methodology, clarity, and novelty.

**Justification For Why Not Higher Score:**

There are areas for improvement. The paper could benefit from a more detailed discussion on the methodology, particularly in terms of model selection and the impact of covariates. Addressing the clarity issues raised by Reviewer XK2E would also strengthen the paper. The concerns about the novelty of the conclusions and the motivation behind certain analyses need to be addressed to establish the paper's contribution more firmly.

**Justification For Why Not Lower Score:**

The paper seems a valuable contribution to the field of CGM research, providing a significant resource for the community. The strengths of the paper lie in its original clinical applications, the comprehensive nature of the dataset collection, and the benchmarking efforts. The inclusion of a code repository enhances the paper's utility and reproducibility.

---

### Decision · Program_Chairs · 2024-01-16

Accept (poster)